# Temporal interpolation of land surface fluxes derived from remote sensing - results with an Unmanned Aerial System

Sheng Wang, Monica Garcia, Andreas Ibrom, Peter Bauer-Gottwein

Department of Environmental Engineering, Technical University of Denmark, 2800 Kgs. Lyngby, Denmark

5  *Correspondence to*: Sheng Wang (shengwang12@gmail.com), Monica Garcia (mgarc@env.dtu.dk)

**Abstract.** Remote sensing imagery can provide snapshots of rapidly changing land surface variables, e.g. evapotranspiration (ET), land surface temperature ($T_s$), net radiation (Rn), soil moisture ($\theta$) and gross primary productivity (GPP), for the time of sensor overpass. However, discontinuous data acquisitions limit the applicability of remote sensing for water resources and ecosystem management. Methods to interpolate between remote sensing snapshot data and to upscale them from 10  instantaneous to daily time scale are needed. We developed a dynamic Soil Vegetation Atmosphere Transfer model to interpolate land surface state variables that change rapidly between remote sensing observations. The Soil-Vegetation, Energy, water and $CO_2$ traNsfer model (SVEN), which combines the snapshot version of the remote sensing Priestley Taylor Jet Propulsion Laboratory ET model and light use efficiency GPP models, incorporates now a dynamic component for the ground heat flux based on the 'force-restore' method and a water balance bucket model to estimate $\theta$ and canopy wetness at 15  a half-hourly time step. A case study was conducted to demonstrate the method using optical and thermal data from an Unmanned Aerial System in a willow plantation flux site (Risoe, Denmark). Based on model parameter calibration with the snapshots of land surface variables at the time of flight, SVEN interpolated UAS based snapshots to continuous records of $T_s$, Rn, $\theta$, ET and GPP for the growing season of 2016 with forcing from continuous climatic data and NDVI. Validation with eddy covariance and other in-situ observations indicates that SVEN can estimate daily land surface fluxes between 20  remote sensing acquisitions with normalized root mean square deviations of the simulated daily $T_s$, Rn, $\theta$, LE and GPP equal to 11.77%, 6.65%, 19.53%, 14.77%, and 12.97%, respectively. This study demonstrates that, in this deciduous tree plantation, temporally sparse optical and thermal remote sensing observations can be used to calibrate soil and vegetation parameters of a simple land surface modelling scheme to estimate "low persistence" or rapidly changing land surface variables with the use of few forcing variables. This approach can also be applied with remotely sensed data from other 25  platforms to fill temporal gaps, e.g. cloud induced data gaps in satellite observation.

## 1 Introduction

Continuous estimates of the coupled exchanges of energy, water and $CO_2$ between the land surface and the atmosphere are essential to understand ecohydrological processes (Jung et al., 2011), to improve agricultural water management (Fisher et al., 2017), and to inform policy decisions for societal applications (Denis et al., 2017). Earth observation (EO) data have

been increasingly used to estimate the land surface-atmosphere flux exchanges at the time of sensor overpass, particularly for regions with scarce ground observations. Optical and thermal remote sensing can provide snapshots of these fluxes such as soil moisture ($\theta$) (Carlson et al., 1995; Sandholt et al., 2002), evapotranspiration (ET) (Fisher et al., 2008; Mu et al., 2013) or gross primary productivity (GPP) (Running et al., 2004) using land surface reflectance or temperature. However, both optical and thermal satellite observations present gaps during cloudy periods, and those gaps may coincide with the time when such information is needed (Westermann et al., 2011), for instance, the prevalence of cloudy weather during the crop growing season in monsoonal regimes (García et al., 2013) and high latitude regions (Wang et al., 2018a). Methods are needed to temporally interpolate and upscale the instantaneous records into continuous daily, monthly or annual estimates (Alfieri et al., 2017; Huang et al., 2016).

As one of the most exciting recent advances in near-Earth observation, Unmanned Aerial Systems (UAS) can favourably fly at a low altitude (< 100-200 m) with flexible revisit times and low cost (Berni et al., 2009; McCabe et al., 2017). Compared to satellites, UAS provide opportunities to acquire high temporal and spatial resolution data under cloudy weather conditions to monitor and understand the surface-atmosphere energy, water and $CO_2$ fluxes (Vivoni et al., 2014). For instance, two-source energy balance models have been extensively applied with UAS thermal imagery for mapping spatial variability of ET in barley fields and vineyard (Hoffmann et al., 2016; Kustas et al., 2018). Zarco-Tejada et al. (2013) applied UAS based hyperspectral and solar-induced fluorescence techniques to infer crop physiological and photosynthesis status in vineyard. Wang et al. (2018b) utilized the vegetation temperature triangle approach with UAS thermal imagery, multispectral imagery and digital surface model to derive high spatial resolution information of root-zone soil moisture for a willow bioenergy site. Wang et al. (2019a) demonstrated the ability of UAS multispectral and thermal imagery for mapping high spatial resolution ecosystem water use efficiency in a willow plantation. However, UAS observations still only provide snapshots of the land surface status at the time of the flight, while conditions such as land surface temperature ($T_s$), net radiation (Rn), $\theta$, ET and GPP between image acquisitions remain unknown.

To continuously estimate land surface-atmosphere energy, water and $CO_2$ fluxes, remote sensing based observations or simulations require either statistical or process-model based approaches to be interpolated into continuous records. The statistical approach is often used to interpolate those land surface variables with high persistence, e.g., which do not change rapidly and can be assumed to be static for several days. For instance, to exclude cloud influence for proxies of vegetation structure e.g. vegetation indices (VI), satellite products use pixel composites to take the maximum value of VI from a given period between 8 and 16 days. To fill the gaps for this period, these 8 or 16 day maximum VI can be statistically interpolated into daily or sub-daily time series data, as the vegetation growth does not change significantly during such a short period. However, the statistical method to interpolate variables that change substantially at sub-daily or daily time scales in response to the surface energy dynamics, e.g. $T_s$, Rn, $\theta$, ET and GPP, could be challenging with low revisit frequency. For instance, Alfieri et al. (2017) found that a return interval of EO observations of no less than 5 days was necessary to statistically interpolate daily ET with relative errors smaller than 20%. To interpolate low persistence variables between remote sensing

acquisitions, a dynamic model based interpolation approach considering the dynamics of the land surface energy balance has great potential.

Ecosystem and land surface models, which can be used to diagnose and predict ecosystem functioning in variable climatic conditions, such as BIOME-BGC (Running and Coughlan, 1988) and Simple Interactive Biosphere Model (SiB2, Sellers et al., 1996), can be used to temporally interpolate the land surface fluxes between EO snapshots with available model drivers and parameter values. Djamai et al. (2016) combined Soil Moisture Ocean Salinity (SMOS) Disaggregation, which is based on the Physical and Theoretical Scale Change (DisPATCh) downscaling algorithm, with the Canadian Land Surface Scheme (CLASS) to temporally interpolate θ at very high spatial and temporal resolutions. Malbéteau et al. (2018) used the ensemble Kalman filter approach to assimilate DisPATCh into a simple dynamic model to temporally interpolate θ. Jin et al. (2018) temporally interpolated AMSR-E based θ estimates with the China Soil Moisture Dataset (SCMD) from the Microwave Data Assimilation system. However, temporal interpolation using complex land surface models requires large data inputs and complicated parameterization schemes. In view of these challenges, simple model-based interpolation can be utilized to interpolate snapshot remote sensing estimates of land surface variables. For instance, using a one-dimensional heat transfer equation, Zhang et al. (2015) interpolated daily $T_s$ on cloudy days. Based on surface energy balance (SEB), Huang et al. (2014) proposed a generic framework with two to twelve parameters to temporally interpolate satellite based instantaneous $T_s$ to diurnal temperatures for clear sky conditions with mean absolute errors from 1.71 to 0.33 °C, respectively. However, model based approaches to temporally interpolate various land surface fluxes such as ET and GPP are rare.

This study aims at developing a simple but operational land surface modeling scheme, which simulates the land surface energy balance and water and $CO_2$ fluxes between the land surface and the atmosphere. We aimed at using prescribed vegetation dynamics from EO based vegetation indices, limited meteorological inputs, and parameters optimized from remote sensing derived fluxes to estimate the temporally continuous land surface variables. It can be used for various conditions even in data-scarce regions by performing parameter calibration with snapshot remote sensing estimates of $T_s$, θ, ET or GPP at the time of overpass. A Soil-Vegetation water and $CO_2$ flux Exchange, eNergy balance model (SVEN) was developed to continuously estimate $T_s$, θ, GPP and ET. The SVEN model is based on a joint ET and GPP model, which combines a light use efficiency GPP model and the Priestley–Taylor Jet Propulsion Laboratory ET model (Wang et al., 2018a). This joint ET and GPP diagnostic model can simulate canopy photosynthesis, evaporation of intercepted water, transpiration and soil evaporation with EO data as inputs. This model serves as a part of the transient surface energy balance scheme, SVEN, which incorporates additional processes and interactions between soil, vegetation and atmosphere, e.g. surface energy balance, sensible heat flux, and θ dynamics, to be able to simulate the land surface fluxes when EO data are not available. Compared to most traditional land surface models, which couple processes of transpiration and $CO_2$ exchange through stomata behaviour and use a 'bottom-up' approach to upscale processes from the leaf scale to the canopy scale (Choudhury and Monteith, 1988; Shuttleworth and Wallace, 1985), SVEN uses a 'top-down' approach to directly simulate water and $CO_2$ fluxes at the canopy scale. SVEN estimates GPP and ET under potential or optimum conditions and then the

potential values are down-regulated by the same biophysical constraints reflecting multiple limitations or stresses. These constraints can be derived from remote sensing and atmospheric data (García et al., 2013; McCallum et al., 2009). In this way, SVEN avoids detailed descriptions and parameterization of complex radiation transfer processes at the leaf level and the scaling process to the canopy level. It maintains a level of complexity comparable to that of operational remote sensing based GPP and ET instantaneous models while being able to predict the fluxes in periods without EO data.

The main objective of this study was to demonstrate a methodology to temporally interpolate sparse snapshot estimates of land surface variables into daily time steps relying on UAS observations. Specific objectives were (1) to develop an operational 'top-down' model to simulate rapidly changed variables e.g. $T_s$, Rn, θ, ET and GPP to interpolate between remote sensing snapshot estimates; (2) to demonstrate the application of this model with UAS observations, calibrating the model with UAS snapshot estimates and forcing it with meteorological data and statistically interpolated VI.

## 2 Study site and data

### 2.1 Study site

This study was conducted in an eddy covariance flux site, Risoe (DK-RCW), which is an 11-hectare willow bioenergy plantation adjacent to the DTU Risoe campus, Zealand, Denmark (55.68°N, 12.11°E), as shown in Figure 1. This site has a temperate maritime climate with the mean annual temperature of about 8.5°C and precipitation of around 600 mm·yr$^{-1}$. The soil texture of this site is loam. The stand consists of two clones ('Inger' and 'Tordis') crossing of *Salix viminalis*, *Salix schwerinii* x *Salix triandra*. In February of 2016, the aboveground parts were harvested following the regular management cycle. Then willow trees grew to a height of approximately 3.5m during the growing season of 2016 (May to October). Rapeseed (*Brassica napus*) was grown in the nearby field. A grass bypass is between the willow plantation and the rapeseed field. An eddy covariance observation system (DK-RCW) has been operated since 2012. Regular UAS flight campaigns with a multispectral camera (MCA, Multispectral Camera Array, Tetracam, Chatsworth, CA, USA) and a thermal infrared camera (FLIR Tau2 324, Wilsonville, OR, USA) onboard were conducted in this site during the growing seasons of 2016. For more details, please refer to Wang et al. (2018b).

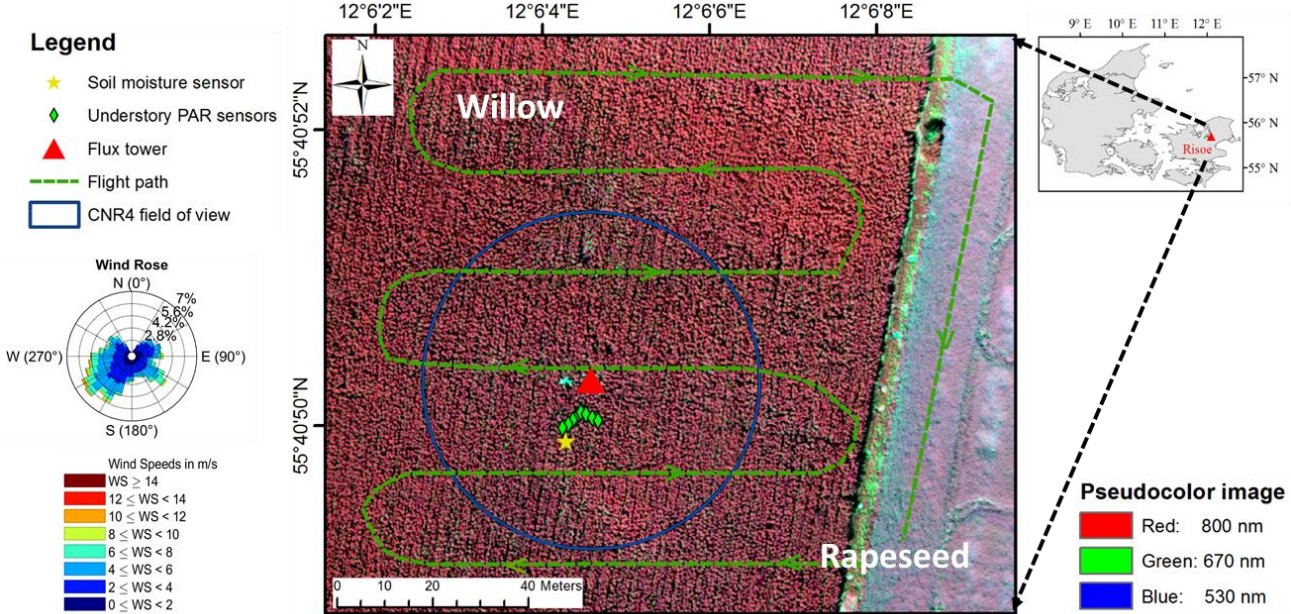

**Figure 1**. Overview of the Risoe willow plantation eddy covariance flux site. The flux tower is the red triangle in the middle of the willow plantation. The green dashed line shows the typical flying path of UAS. Green diamonds indicate the location of the understory PAR sensors. The yellow star refers to the soil moisture sensor. The blue circle indicates the CNR4 field of view. The wind rose refers to the wind direction and frequency in 2016. The base map is a multispectral pseudo-colour image collected on August 1st, 2016 with 800, 670 and 530 nm as red, green and blue channels, respectively.

## 2.2 Data

In-situ data used in this study include standard eddy covariance and micrometeorological observations, such as GPP, ET, Rn, incoming longwave radiation ($LW_{in}$), outgoing longwave radiation ($LW_{out}$) and incoming shortwave radiation ($SW_{in}$), air temperature ($T_a$), vapor pressure deficit (VPD) and θ. These meteorological variables were measured at the height of 10 m above the ground. Meanwhile, the $CO_2$ and water vapor eddy covariance system was adjusted to around 2 m above the maximum canopy height. The eddy covariance data processing followed the same procedures as in Pilegaard et al. (2011), Ibrom et al. (2007) and Fratini et al. (2012), i.e. the standard ICOS processing method. The raw data were aggregated into half-hourly records. The flux partitioning to separate GPP and respiration was done by the look-up table approach (Reichstein et al., 2005) based on the R-package REddyProc (Wutzler et al., 2018) with the half-hourly net ecosystem exchange, $T_a$ and $SW_{in}$ as inputs.

A UAS equipped with MCA and FLIR cameras was used to collect the Normalized Difference Vegetation Index (NDVI) and land surface temperature ($T_s$) (Wang et al., 2019). For each flight campaign, the digital surface model (DSM), multispectral reflectance and thermal infrared orthophotos were generated. For details on the UAS, sensors and image processing, refer to

Wang et al. (2018b). To continuously estimate the land surface fluxes from UAS, the collected mean NDVI for the willow patch was temporally statistically interpolated into half-hour continuous records by the Catmull-Rom spline method (Catmull and Rom, 1974). The interpolated NDVI was converted into the fraction of intercepted photosynthetically active radiation ($f_{IPAR}$), which can also be assumed equal to the fraction of vegetation cover based on Fisher et al. (2008). The canopy height

5   $h_c$ was obtained from the DSM generated from RGB images and then was statistically interpolated into the continuous half-hourly record based on in-situ $f_{IPAR}$. The collected $T_s$ and NDVI from UAS were used to estimate θ based on the modified temperature-vegetation triangle approach as Wang et al. (2018b). Values of the observed NDVI, $T_s$ and the estimated θ from each UAS flight campaign are shown in Table 1. The statistically interpolated NDVI and $h_c$ were used as model inputs/forcing.

10   As technical issues, parts of UAS data on June 24[th] and August 1[st] were missing (Table 1) and in-situ measurements were used to represent these missing values. For instance, to fill a prolonged gap for UAS observations in June of 2016 and simulate the growth process of willow trees, in-situ observations were added to June 24[th]. For model calibration, the instantaneous values of the $T_s$ and θ estimated from the seven UAS flights were used as reference. The seven UAS flights resulted in an average frequency of 25 days for this growing season. The minimum revisit time was 10 days in the willow

15   emerging period between May 2[nd] and May 12[th]. The maximum revisit time was 67 days between August 1[st] to October 7[th] when the willow canopy was dense and stable.

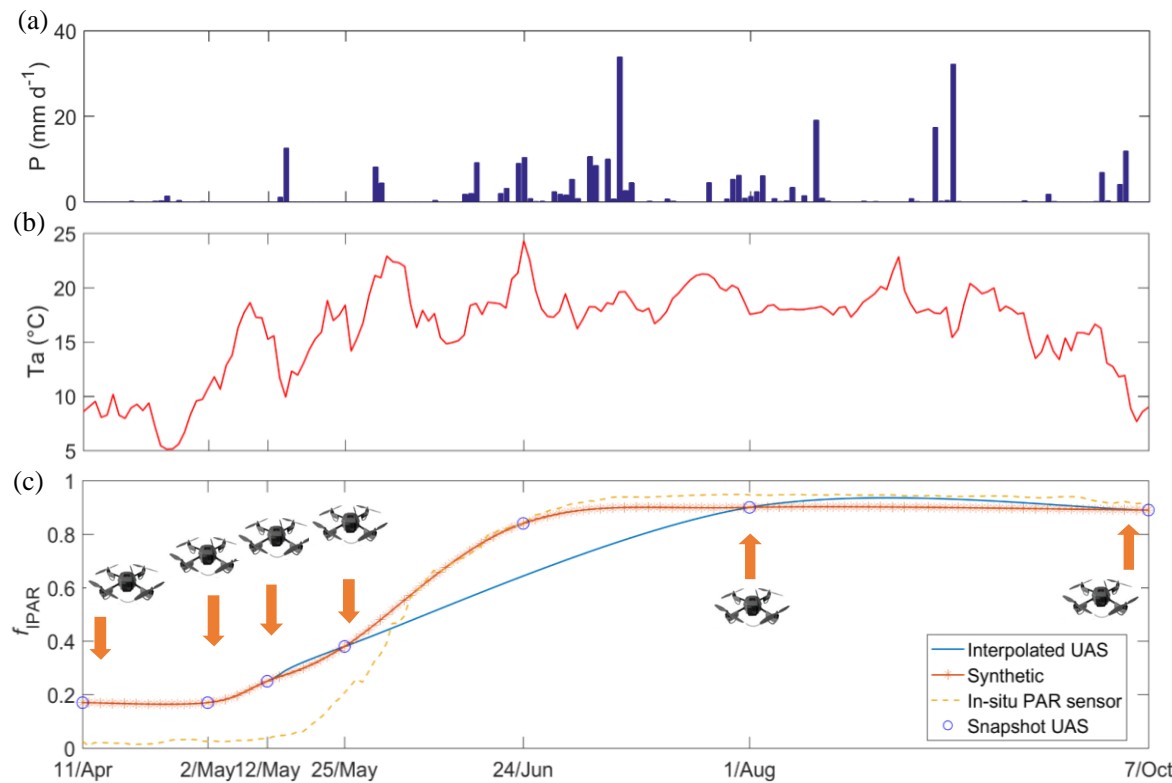

**Figure 2.** (a) Daily precipitation (P, mm·d$^{-1}$), (b) Daily air temperature (T$_a$, °C), and (c) Daily fraction of the intercepted PAR ($f_{IPAR}$) interpolated from UAS based NDVI during the growing season of 2016.

**Table 1.** NDVI, surface temperature and soil moisture information from UAS and in-situ data. * indicates that no available data from UAS due to technical issues and in-situ data were used to represent UAS snapshots. $f_{IPAR}$ is the fraction of intercepted PAR. T$_s$ is land surface temperature (°C). θ is volumetric soil moisture (m$^3$·m$^{-3}$). For methods of θ estimation and detailed weather conditions, please refer to Wang et al. (2019b).

| Date | Acquisition time | Weather | $f_{IPAR}$ UAS | $f_{IPAR}$ obs | T$_s$ UAS | T$_s$ obs | θ UAS | θ obs | Growth stage |
|---|---|---|---|---|---|---|---|---|---|
| 11-Apr-16 | 11:13-11:26 | Cloudy | 0.22 | 0.03 | 14.98 | 15.95 | 0.27 | 0.28 | Early growth |
| 2-May-16 | 14:40-14:55 | Cloudy | 0.22 | 0.03 | 18.29 | 19.13 | 0.27 | 0.30 | Early growth |
| 12-May-16 | 10:44-11:55 | Sunny | 0.3 | 0.04 | 24.84 | 23.57 | 0.25 | 0.27 | Early growth |
| 25-May-16 | 10:11-10:23 | Sunny | 0.43 | 0.20 | 28.08 | 28.31 | 0.26 | 0.26 | Early growth |
| 24-Jun-16 | 12:00-12:30 | Sunny | 0.84* | 0.84 | 26.60* | 26.60 | 0.21* | 0.21 | Dense vegetation |
| 1-Aug-16 | 10:06-10:14 | Cloudy | 0.95 | 0.95 | 18.33* | 18.33 | 0.20* | 0.20 | Dense vegetation |
| 7-Oct-16 | 11:41-11:55 | Sunny | 0.94 | 0.91 | 11.10 | 10.41 | 0.16 | 0.19 | Dense vegetation |

## 3 Method

The SVEN model is an operational and parsimonious remote sensing based land surface modeling scheme expanding the capabilities of the remote sensing GPP and PT JPL-ET model (Wang et al., 2018a) to be dynamic. It runs at half-hourly time steps and can temporally interpolate the instantaneous land surface variables, such as T$_s$, Rn, θ, ET and GPP, into continuous records.

### 3.1 Model description

SVEN consists of a surface energy balance module, a water balance module and a CO$_2$ flux module. In the energy balance module, SVEN estimates the surface temperature and ground heat flux relying on the land surface energy balance equations and the 'force-restore' method (Noilhan and Mahfouf, 1996; Noilhan and Planton, 1989) to consider the energy exchange between ground and soil/vegetation on surface. The water balance module includes the Priestley–Taylor Jet Propulsion Laboratory (PT-JPL) model for ET estimation and a simple 'bucket' model representing the upper soil column to simulate soil water dynamics and runoff generation. The CO$_2$ flux module uses a light use efficiency (LUE) model for GPP estimation, which is connected to ET via the same canopy biophysical constraints. Figure 3 shows the major processes simulated in SVEN. Detailed information on these three modules is outlined below.

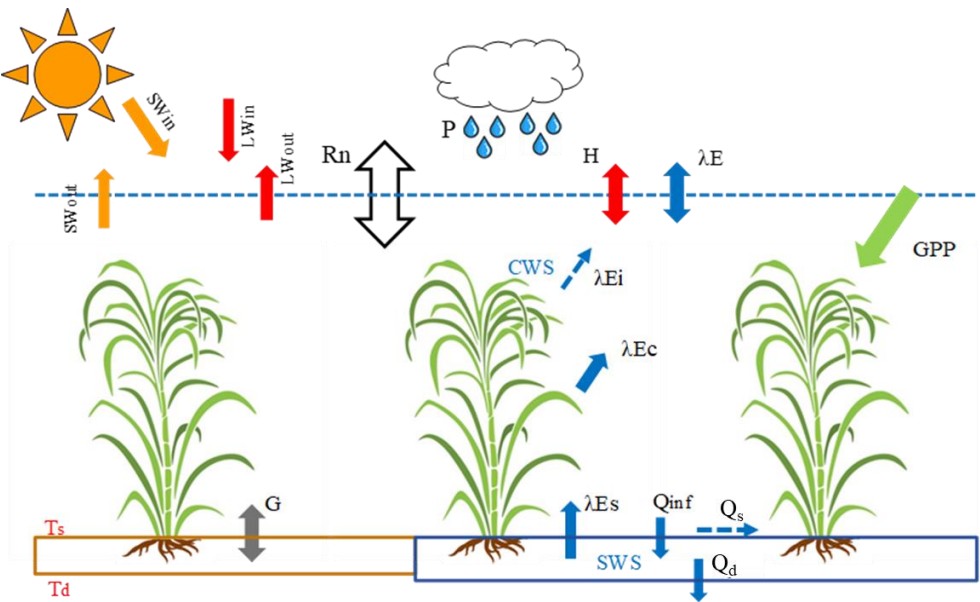

**Figure 3**. Major land surface processes simulated in SVEN. These processes include land surface energy balance, water fluxes and $CO_2$ assimilation ($SW_{in}$: incoming shortwave radiation; $SW_{out}$: outgoing shortwave radiation; $LW_{in}$: incoming longwave radiation; $LW_{out}$: outgoing longwave radiation; Rn: net radiation; G: ground heat flux; $T_s$: the surface temperature; $T_d$: the deep soil temperature; H: sensible heat flux; P: precipitation; λE: latent heat flux; λEi: latent heat flux of the intercepted water; λEc: latent heat flux of transpiration; λEs: latent heat flux of soil evaporation; CWS: canopy water storage; SWS: soil water storage; $Q_{inf}$: infiltration; $Q_d$: drainage; $Q_s$: surface runoff; GPP: gross primary productivity.

### 3.1.1 Surface Energy Balance Module

The instantaneous net radiation is estimated based on the surface energy balance, as shown in Eq. (1). The surface emissivity is approximated according to an empirical relation with NDVI as Eq. (2) (Van de Griend and M.Owe., 1993). The surface albedo (A) is estimated from the simple ratio vegetation index (SR) and it shows that albedo generally decreases as vegetation greenness increases as Eq. (3 and 4) (Gao, 1995).

$$Rn = (1 - A)SW_{in} + (1 - \varepsilon)LW_{in} - \varepsilon\sigma T_s^4 \qquad (1)$$

$$\varepsilon = \begin{cases} 0.986 & (NDVI > 0.608) \\ 1.0094 + 0.047 \cdot \ln(NDVI) & (0.131 < NDVI < 0.608) \\ 0.914 & (NDVI < 0.131) \end{cases} \qquad (2)$$

$$A = 0.28 - 0.14e^{(-6.08/SR^2)} \qquad (3)$$

$$SR = (1 + NDVI)/(1 - NDVI) \qquad (4)$$

Where Rn is the instantaneous net radiation (W·m⁻²). $SW_{in}$ is the instantaneous incoming shortwave radiation (W·m⁻²). $LW_{in}$ is the instantaneous incoming longwave radiation (W·m⁻²). σ is the Stefan-Boltzmann constant ($5.670367 \times 10^{-8}$ W·m⁻²·K⁻⁴).

At the surface, Rn is dissipated as latent, sensible and ground heat fluxes, as Eq. (5). The latent heat flux is estimated from the PT-JPL ET model and the sensible heat flux, H, is calculated based on the temperature gradient between the surface and air and a bulk aerodynamic resistance. The instantaneous ground heat flux G is estimated from the ´force-restore´ method (Noilhan and Planton, 1989).

$$\frac{dS}{dt} = SW_{in} - SW_{out} + LW_{in} - LW_{out} - H - \lambda E - G \tag{5}$$

Where $\frac{dS}{dt}$ is the heat storage change over time (W·m⁻²). SW is shortwave radiation (W·m⁻²) and LW is longwave radiation (W·m⁻²). The subscripts $_{in}$ and $_{out}$ refer to incoming and outgoing, respectively. λE represents the latent heat flux (W·m⁻²). H refers to the sensible heat flux (W·m⁻²). G is the ground heat flux (W·m⁻²).

The surface temperature was estimated by the 'force-restore' method, which considers two opposite effects on surface temperature variabilities, as shown in Eq. (6). The first term $(R_n - \lambda E - H)$ represents the forcing from the surface-atmosphere interface. The second term $(T_s - T_d)$ is the gradient between the surface temperature and deep soil temperature. It indicates the tendency from the deep soil to restore $T_s$ (responding to surface energy forcing) to the $T_d$ value, which is more stable over time.

$$\frac{dT_s}{dt} = C_T(R_n - \lambda E - H) - C_d(T_s - T_d) \tag{6}$$

$$\frac{dT_d}{dt} = \omega(T_s - T_d) \tag{7}$$

$$\frac{1}{C_T} = \frac{1-f_c}{C_{sat}(\frac{SWS_{max}}{SWS})^{\frac{b}{2\ln(10)}}} + \frac{f_c}{C_{veg}} \tag{8}$$

$$C_d = 2\pi\omega \tag{9}$$

Where $T_s$ is the land surface temperature (°C). $T_d$ refers to the deep soil temperature (°C) calculated by applying a low-pass filter to $T_s$ with the cut-off frequency of 24 hours. ω is the frequency of oscillation 1/24 (h⁻¹). $C_T$ is a force-restore thermal coefficient for the surface heat transfer (K·m²·J⁻¹) and is influenced by the effective relative θ. $C_{sat}$ is the force-restore thermal coefficient for saturated soil (K·m²·J⁻¹). The parameter b is the slope of the retention curve for the force-restore thermal coefficient. $C_{veg}$ is the force-restore thermal coefficient for vegetation (K·m²·J⁻¹). $f_c$ is the fractional cover of vegetation and is assumed equal to $f_{IPAR}$ as the supplemental Table S1 (Fisher et al., 2008). $SWS_{max}$ is the maximum soil water storage (m) and SWS is the actual one (m). $C_d$ is diurnal periodicity based on ω (h⁻¹).

The sensible heat flux, H, is estimated based on the temperature gradient between the surface and air, as shown in Eq. (10).

$$H = \rho c_p (T_s - T_a)/r_a \qquad (10)$$

Where $\rho$ is the air density (kg·m$^{-3}$). $c_p$ is the specific heat capacity of air (J·kg$^{-1}$·K$^{-1}$). $T_s$ is the land surface temperature (°C). $T_a$ is the air temperature (°C). $r_a$ is the aerodynamic resistance for heat transfer (s·m$^{-1}$).

Aerodynamic resistance to turbulent transport under neutral conditions ($r_{aN}$) can be expressed as Eq. (11) (Brutsaert, 1982).

$$r_{aN} = \frac{\ln\left(\frac{z-d}{z_{0m}}\right)\ln\left(\frac{z-d}{z_{0h}}\right)}{k^2 u} \qquad (11)$$

$$d = 0.67 h_c \qquad (12)$$

$$z_{om} = 0.1 h_c \qquad (13)$$

$$z_{oh} = \frac{z_{0m}}{e^{kB-1}} \qquad (14)$$

Where $h_c$ is the canopy height (m). The parameter d is the zero displacement height (m) and z is the velocity reference height (m). $z_{om}$ is the aerodynamic roughness length for momentum (m). $z_{oh}$ is the aerodynamic roughness length for the heat transfer (m). u is the horizontal wind velocity at reference height (m·s$^{-1}$). kB$^{-1}$ is a parameter to account for the difference between the aerodynamic and radiometric temperatures and a constant value of 2.3 is adopted in this study (Garratt and Hicks, 1973). k is the von Karman constant (0.4).

The aerodynamic resistance is corrected for the atmospheric stability as shown in Eq. (15) (Huning and Margulis, 2015). $\Psi_m$ is the stability correction factor for momentum. $\Psi_h$ is the stability correction factor for sensible heat flux. For unstable conditions (negative temperature gradient), the stability correction factors are less than 1.0 and the correction reduces the resistance and enhances turbulence, while for stable conditions they are greater than 1.0 and the correction increases the resistance and suppresses turbulence.

$$r_a = r_{aN}\Psi_m\Psi_h \qquad (15)$$

When the atmospheric condition is unstable ($R_{iB} \leq 0$), $\Psi_m$ and $\Psi_h$ are estimated from Eq. (16).

$$\Psi_h = \Psi_m{}^2 = (1 - 15 R_{iB})^{-1/2} \qquad (16)$$

When atmospheric condition is stable ($0 \leq R_{iB} < 0.2$), $\Psi_m$ and $\Psi_h$ are estimated from Eq. (17).

$$\Psi_h = \Psi_m = (1 - 5 R_{iB})^{-1} \qquad (17)$$

$$R_{iB} = \frac{\left(\frac{g}{T_s}\right)\partial T_s/\partial z}{\left(\frac{\partial u}{\partial z}\right)^2} \qquad (18)$$

Where $R_{iB}$ is the bulk Richardson number, g is the gravitational acceleration.

### 3.1.2 Water balance module

The water balance module simulates evaporation of intercepted water, plant transpiration, soil evaporation, soil water infiltration and drainage. The evapotranspiration is estimated based on a modified PT-JPL ET model (Wang et al., 2018a). The PT-JPL ET model has been demonstrated as one of best-performing global remote sensing ET algorithms (Chen et al., 2014; Ershadi et al., 2014; Miralles et al., 2016; Vinukollu et al., 2011). Thus, it was selected for ET estimation. The PT-JPL model (Fisher et al., 2008) uses the Priestley-Taylor (1972) equation to calculate the potential evapotranspiration, and then incorporates eco-physiological variables to down-regulate potential evapotranspiration to actual evapotranspiration. PT-JPL is a three-source evapotranspiration model to simulate evaporation of intercepted water (Ei), transpiration (Ec) and soil evaporation (Es) as following equations.

$$\lambda ET = \lambda Ei + \lambda Ec + \lambda Es \tag{19}$$

$$\lambda Ei = f_{wet} \cdot \alpha \Delta/(\Delta + \gamma) \cdot Rnc \tag{20}$$

$$\lambda Ec = (1 - f_{wet}) \cdot f_g \cdot f_M \cdot f_{Ta} \cdot \alpha_c \Delta/(\Delta + \gamma) \cdot Rnc \tag{21}$$

$$\lambda Es = f_\theta \cdot \alpha \Delta/(\Delta + \gamma) \cdot (Rns - G) \tag{22}$$

Where $\lambda ET$ is the latent heat flux for total evapotranspiration (W·m$^{-2}$), $\lambda Ei$ is the latent heat flux due to evaporation of intercepted water (W·m$^{-2}$), $\lambda Ec$ is the latent heat flux due to transpiration (W·m$^{-2}$), and $\lambda Es$ is the latent heat flux due to evaporation of soil water (W·m$^{-2}$). The quantity $f_{wet}$ is the relative surface wetness to partition the evapotranspiration from the intercepted water and canopy transpiration. $f_g$ is the green canopy fraction indicating the proportion of active canopy. $f_M$ is the plant moisture constraint. $f_{Ta}$ is the plant temperature constraint reflecting the temperature limitation of photosynthesis. $f_\theta$ is the θ constraint. These constraints vary from 0 to 1 to account for the relative reduction of potential $\lambda ET$ under limiting environmental conditions. Rnc and Rns are the net radiation for canopy and soil, respectively. The partitioning of PAR and net radiation between canopy and soil is calculated following the Beer-Lambert law (Supplemental Table S1). G is the ground heat flux. $\Delta$ is the slope of saturation vapor pressure versus temperature curve. $\gamma$ is the psychrometric constant. $\alpha$ is an empirical ratio of potential evapotranspiration to equilibrium potential evapotranspiration (Priestley-Taylor coefficient). The suggested value for $\alpha$ is 1.26 in the PT-JPL model (Fisher et al., 2008).

In the original model, $f_{wet}$ was estimated from air relative humidity (Fisher et al., 2008). In this study, $f_{wet}$ is modified to be defined as a ratio between the actual canopy water storage (CWS) and the maximum canopy water storage (CWS$_{max}$) as Eq. (23) (Noilhan and Planton, 1989). CWS is the amount of intercepted water and CWS$_{max}$ is the maximum possible amount of intercepted water (mm), taken as 0.2LAI kg·m$^{-2}$ (Dickinson, 1984). $f_{wet}$ depends on both the precipitation rate and LAI, which is more reasonable than only depending on air relative humidity in the original model.

$$f_{wet} = \frac{CWS}{CWS_{max}} \tag{23}$$

In this study, we determined CWS with a prognostic equation (24) with the constraint that CWS is smaller than $CWS_{max}$.

$$\frac{dCWS}{dt} = f_c \cdot P - Ei \tag{24}$$

Where $f_c$ is the fraction of vegetation cover and here it is assumed to be equal to $f_{IPAR}$ (Fisher et al., 2008). P and Ei are the rainfall rates and evaporation from the intercepted water, respectively ($m \cdot s^{-1}$).

The effective precipitation rate is estimated as the residual of the rainfall rate and change of CWS as Eq. (25).

$$P_e = P - dCWS \tag{25}$$

To simulate the dynamics of water storage in the soil, SVEN uses a simple 'bucket' model. Here the infiltration rate ($Q_{inf}$) is equal to the effective rainfall rate ($P_e$), when the soil water is not saturated. Thus, SWS is calculated based on a prognostic equation with a constraint that SWS is smaller than $SWS_{max}$.

$$Q_{inf} = P_e \tag{26}$$

$$\frac{dSWS}{dt} = Q_{inf} - Ec - Es - Q_d \tag{27}$$

When soil water is saturated, SWS is equal to $SWS_{max}$ and surface runoff ($Q_s$) occurs as Eq. 29.

$$Q_{inf} = Ec + Es + Q_d \tag{28}$$

$$Q_s = P_e - Q_{inf} \tag{29}$$

Where SWS is soil water storage (m). $P_e$, $Ec$, $Es$, $Q_d$ and $Q_s$ are the effective rainfall rates, transpiration rates, evapotranspiration rates from soil, drainage rates and surface runoff ($m \cdot s^{-1}$), respectively.

Soil water drainage, which is leakage out of the lower boundary of the flow domain (Romano et al., 2011), is computed by assuming the condition of a unit gradient of the total hydraulic potential at the lowest boundary and using the van Genuchten (1980) soil-water retention relationship as Eq. (30).

$$Q_d = K_s \sqrt{\theta_e} (1 - (1 - \theta_e^{1/(1-1/n)})^{1-1/n})^2 \tag{30}$$

$$\theta_e = \frac{\theta - \theta_r}{\theta_s - \theta_r} \tag{31}$$

Where $K_s$ is the saturated hydraulic conductivity ($m \cdot s^{-1}$). n is the shape parameter of the van Genuchen (1980) soil-water retention relationship, and depends on the pore-size distribution. $\theta$ is the volumetric soil moisture ($m^3 \cdot m^{-3}$). $\theta_e$ is the effective soil moisture ($m^3 \cdot m^{-3}$). $\theta_s$ is the saturated soil moisture ($m^3 \cdot m^{-3}$). $\theta_r$ is the residual soil moisture ($m^3 \cdot m^{-3}$).

### 3.1.3 $CO_2$ flux module

The photosynthesis in the $CO_2$ flux module is calculated from a modified light use efficiency (LUE) model (Wang et al., 2018a) linked to the biophysical constraints for canopy transpiration of the PT-JPL model. The LUE GPP model is a robust and widely used method to estimate GPP across various ecosystems and climate regimes (McCallum et al., 2009). The LUE models, e.g. CASA (Potter et al., 1993) or the MODIS algorithm (Running et al., 2004), are based on the assumption that plants optimize canopy LUE or whole canopy carbon gain per total PAR absorbed as originally suggested by (Monteith, 1972) for net primary productivity. The formula of the LUE GPP model used in this study is shown in Eq. (32) and it is partly based on the Carnegie-Ames-Stanford-Approach model (Potter et al., 1993) with modification to include an additional constraint accounting for the fraction of the canopy that is photosynthetically active (Fisher et al., 2008). Other constraints such as thermal regulation (Wang et al., 2018a) reflect changes in LUE due to environmental factors and are the same for regulating ETc (Eq. 21).

$$GPP = LUE_{max} \cdot PARc \cdot f_g \cdot f_M \cdot f_{Ta} \cdot f_{VPD} \tag{32}$$

Where $LUE_{max}$ is the maximum LUE ($g \cdot C \cdot MJ^{-1}$). PARc is the daily photosynthetically active radiation (PAR) ($MJ \cdot m^{-2} \cdot d^{-1}$) intercepted by the canopy and it is calculated based on the extinction of PAR within the canopy using the Beer-Lambert law (Supplemental Table S1). $f_g$ is the green canopy fraction indicating the proportion of active canopy. $f_M$ is the plant moisture constraint. $f_{Ta}$ is the air temperature constraint reflecting the temperature limitation of photosynthesis. $f_{VPD}$ is the VPD constraint reflecting the stomatal response to the atmospheric water saturation deficit. All these constraints range from 0 and 1 and represent the reduction of maximum GPP under limiting environmental conditions. For more details, please refer to the supplemental Table S1.

### 3.2 Model implementation

The SVEN model requires shortwave incoming ($SW_{in}$), longwave incoming ($LW_{in}$), air temperature ($T_a$), air pressure ($P_s$), relative humidity (RH), wind speed (u), precipitation (P), canopy height (z), and vegetation information (NDVI) as inputs (Supplemental Table S2). The model inputs of this study were obtained from meteorological data, UAS derived observations or estimates. The simulation outputs of this model are shown in Supplemental Table S4. The initial conditions for the model include an initial canopy water storage ($CWS_{in}$), an initial soil water storage ($SWS_{in}$), initial surface temperature ($T_{s0}$) and initial deep soil temperature ($T_{d0}$) as shown in Supplemental Table S3. The initial conditions to run the model (11-April-2016 to 7-October-2016) were obtained by performing spin-up simulations from 11-March-2016 to 11-April-2016. The details of model implementation are shown in Figure 4.

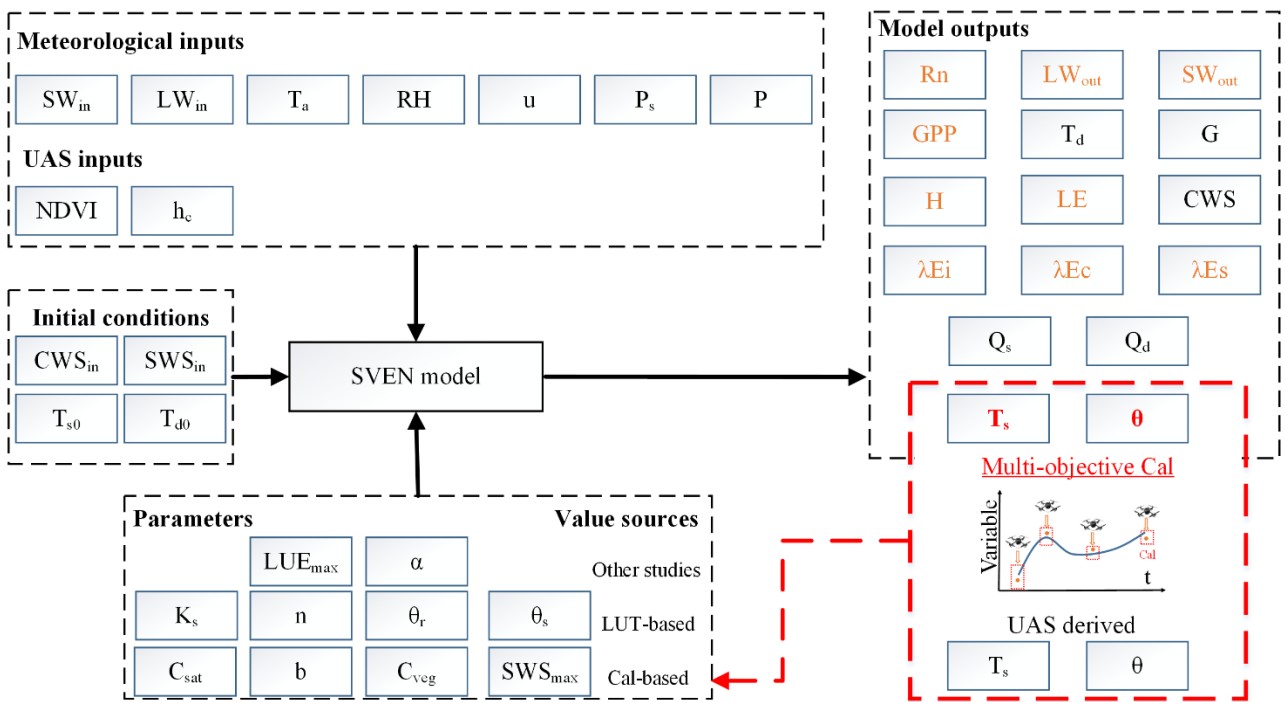

**Figure 4**. Model implementation of this study. UAS and meteorological data were used as inputs of the SVEN model. Values of the SVEN parameters were obtained from other studies, look-up tables (LUT-based), or model calibration with UAS derived variables (Cal-based). In the model outputs, variables with the highlighted red colour ($T_s$ and θ) refers to the variables calibrated with UAS derived observations or estimates. The red shaded box refers represents the multi-objective calibration process with UAS derived $T_s$ and θ. The variables with orange colours are retrievable from remote sensing techniques.

The SVEN model has six parameters mostly related to physical soil properties for heat transfer and infiltration (Table 2). The parameter values can be obtained from multiple approaches including look-up tables based on soil texture, parameter values of the similar biome or soil types in other studies, field measurements, or model parameter optimization with in-situ measurements or with remote sensing data. This study used a combination of these approaches to obtain model parameter values (Figure 4). The parameters, for instance maximum light use efficiency ($LUE_{max}$), to drive the snapshot version of SVEN are obtained from the nearby similar deciduous temperate forest ecosystem (Wang et al., 2018a). The shape parameter of the van Genuchen (1980) soil-water retention relationship (n) and the saturated hydraulic conductivity ($K_s$) were obtained from a look-up table (Carsel and Parrish, 1988). The values for loamy soil as shown in the supplemental Table S5 were used, according to the soil texture of this site. The rest of parameters related with soil and vegetation physical properties, $C_{sat}$, b, $C_{veg}$ and $SWS_{max}$, were obtained by calibrating models with instantaneous $T_s$ and θ from seven UAS flight campaigns (Table 1) rather than calibration with in-situ measurements of ET or GPP (e.g. eddy covariance data) as in other studies. Calibrating

the model with the remotely sensed instantaneous estimates instead of ground measurements facilitates the application of this approach to data-scarce regions. The calibration of $C_{sat}$, b, $C_{veg}$ and $SWS_{max}$ was conducted using the Monte Carlo optimization. The parameter values were sampled 20,000 times with a uniform distribution and the corresponding parameter ranges as shown in Table 2. The objective function for optimization is the root mean square deviation (RMSD) between the observed and simulated values. With two objective functions for $T_s$ and $\theta$ respectively, the multiple objective optimization method (Pareto front) as Yapo et al. (1998) was used to identify the optimized parameter values.

**Table 2**. Information on the model parameters of SVEN and their ranges for all soil or biome types

| Parameters | Description | Unit | Range | Reference | Source for this study |
|---|---|---|---|---|---|
| $LUE_{max}$ | Maximum light use efficiency | $g \cdot C \cdot m^{-2} \cdot MJ^{-1}$ | 0-5 | Wang et al. (2018a) | Other studies |
| $\alpha$ | Priestley-Taylor coefficient | [-] | 1-3 | Fisher et al. (2018) | Other studies |
| $C_{sat}$ | The force-restore thermal coefficient for saturated soil | $10^{-6} K \cdot m^2 \cdot J^{-1}$ | [3, 15] | Noilhan and Planton (1989) | Model calibration |
| b | The slope of the retention curve for the force-restore thermal coefficient | [-] | [4.05, 11.4] | Noilhan and Planton (1989) | Model calibration |
| $C_{veg}$ | The force-restore thermal coefficient for vegetated surface | $10^{-6} K \cdot m^2 \cdot J^{-1}$ | [1, 10] | Calvet et al. (1998) | Model calibration |
| $SWS_{max}$ | Maximum soil water storage | m | [0, 1] | Boegh et al. (2009) | Model calibration |
| $K_s$ | The saturated hydraulic conductivity | $mm \cdot h^{-1}$ | [0.05, $50.0 \times 10^3$] | Dettmann et al. (2014) | Look-up table |
| n | The shape parameter of the van Genuchen (1980) soil-water retention relationship | \ | [1.01, 2.5] | Dettmann et al. (2014) | Look-up table |
| $\theta_s$ | Saturated soil moisture | $m^3 \cdot m^{-3}$ | [0.36, 0.46] | Carsel and Parrish (1988) | Look-up table |
| $\theta_r$ | Residual soil moisture | $m^3 \cdot m^{-3}$ | [0.034, 0.100] | Carsel and Parrish (1988) | Look-up table |

**3.3 Model assessment**

We used independent eddy covariance data to validate model outputs. However, due to the energy balance closure issue (Wilson et al., 2002), the sum of sensible heat (H) and latent heat (LE) as measured by the eddy covariance method is generally not equal to the available energy (net radiation minus ground heat flux, $R_n - G$). This study used the Bowen ratio approach to correct energy balance closure errors of eddy covariance data. Using the ratio of 30 min sensible heat to ET (Bowen ratio), LE measurements can be corrected as follows (Twine et al., 2000). The LE data with the 30 min energy balance closure error larger than 20% were excluded in the validation.

$$LE = \frac{R_n - G}{H\_EC\_raw + LE\_EC\_raw} LE\_EC\_raw \tag{33}$$

Where LE is corrected latent heat by assuming the constant Bowen ratio ($W \cdot m^{-2}$). $R_n$ is net radiation ($W \cdot m^{-2}$). G is ground heat flux ($W \cdot m^{-2}$). H_EC_raw is uncorrected sensible heat ($W \cdot m^{-2}$) and LE_EC_raw is uncorrected latent heat ($W \cdot m^{-2}$).

The SVEN model was developed to interpolate between remote sensing data acquisitions and to produce continuous daily records. Thus, the observed $T_s$, Rn, LE and GPP are from the eddy covariance system and the in-situ $\theta$ measurements at the depth of 15 cm (sensor location in Figure 1) were used to validate the simulated variables at the daily time scale. Statistics including RMSD, the coefficient of determination ($R^2$), relative errors (RE) and normalized RMSD (NRMSD, the ratio between RMSD and the range of observations) were used in validation.

We also analyzed how the model skill changes depending on vegetation cover and overcast (diffuse radiation) conditions by looking at model residuals as typically remote sensing models may be biased to sunny conditions. Scatterplots between model residuals and NDVI and the diffuse radiation fraction were examined. As the ratio between the actual ($SW_{in}$) and potential ($SW_{in}pot$) can be the indicator of the diffuse radiation fraction (Wang et al., 2018a), we used this ratio to indicate the diffuse radiation fraction. This analysis can help to understand possible methods to improve the SVEN model. To check the capability of the SVEN model to interpolate half-hourly and monthly time series fluxes, the simulated land surface variables were also validated at half-hourly and monthly time scales, in addition to the daily time scale.

## 4 Results and discussion

### 4.1 Model parameter estimation

Figure 5 illustrates the results of model parameter calibration with UAS derived snapshot $\theta$ and $T_s$ (Table 1). With RMSDs of $\theta$ and $T_s$ as objective functions, a significant trade-off between the performance of $\theta$ and $T_s$ simulations is observed as a Pareto front (the red curve) in Figure 4. The x-axis shows the performance of simulating $\theta$. The smaller the RMSD values are, the better the model performance with respect to this variable. The minimum, however, lies in a range, where the model performance of the other variable, $T_s$, is highest (y-axis). From the viewpoint of multi-objective optimization, the solutions at the Pareto front are equally good. By considering RMSDs of $T_s$ less than 2 °C and RMSDs of $\theta$ as small as possible, we selected the point close to the red arrow of Figure 4, which corresponds to the RMSDs of $\theta$ and $T_s$ equal to 2.99% $m^3 \cdot m^{-3}$ and 1.92 °C, respectively. The values of $C_{sat}$, b, $C_{veg}$ and $SWS_{max}$ at this Pareto front point are equal to $6.94 \times 10^{-6}$ $K \cdot m^2 \cdot J^{-1}$, 5.20, $2.18 \times 10^{-6}$ $K \cdot m^2 \cdot J^{-1}$ and $5.54 \times 10^{-1}$ m, respectively. Furthermore, we also analysed the variability of optimized parameter values as shown in supplementary Figure S1. $C_{veg}$ and $SWS_{max}$ show low variation of coefficients (CVs), and this indicates the parsimony of the SVEN model. Meanwhile, $C_{sat}$ and b show relatively higher CVs. This may be due to equifinality between $C_{sat}$ and b, which relate to soil thermal properties (Eq. 8) and could compensate each other. Notably, these calibrated values, e.g. $SWS_{max}$, represents the equivalent calibrated parameter value and might be different from the actual physical conditions.

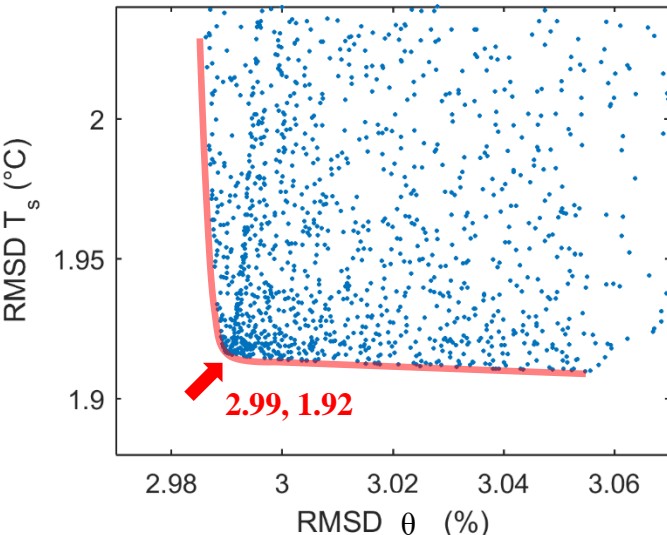

**Figure 5**. Objective function values of evaluated parameter sets and corresponding Pareto front. The x-axis is the objective function for simulating $\theta$. The y-axis is the objective function for simulating $T_s$. Each dot corresponds to one simulation performance. Each of the simulations represents a different combination of candidate parameter sets. The dot closest to the
red arrow is chosen to be the optimal parameter set for SVEN continuous simulation. $C_{sat}$, b, $C_{veg}$ and $SWS_{max}$ at the Pareto front point are $6.94 \times 10^{-6}$ K·m²·J⁻¹, 5.20, $2.18 \times 10^{-6}$ K·m²·J⁻¹ and $5.54 \times 10^{-1}$ m , respectively.

## 4.2 Validation at the daily time scale

Figure 6 shows the time-series data of the interpolated daily $T_s$, Rn, $\theta$, LE and GPP and their validation. The simulated daily $T_s$, Rn, $\theta$, LE and GPP capture well with the observed temporal dynamics of land surface variables at this site. $R^2$ for daily
$T_s$, Rn, $\theta$, LE and GPP are 0.90, 0.92, 0.50, 0.70 and 0.79, respectively. RMSDs for the simulated daily $T_s$, Rn, $\theta$, LE and GPP are 2.35 °C, 14.49 W·m⁻², 1.98% m³·m⁻³, 16.62 W·m⁻² and 3.01 g·C·m⁻²·d⁻¹, respectively. Such simulation accuracy demonstrates that SVEN is capable of temporal interpolating the snapshot estimates or observations between remote sensing acquisitions to form continuous daily records.

For the simulated $T_s$, during the early growth stage (before June), the SVEN model simulated quite accurately the temporal
dynamics. However, during the dense vegetation stage (high NDVI), the model generally tends to overestimate $T_s$. Similarly, SVEN underestimated Rn during the early growth stage, but overestimated Rn for the dense vegetation stage. These biases can also be identified from the boxplots of model residuals and NDVI (Fig. 7b), which shows that the model underestimates Rn in low NDVI and vice-versa. One of the reasons for this error could be the uncertainty in the estimated surface albedo. The albedo in the SVEN model was determined by the simple empirical formula as Eq. (3) with a high value in the early
growth stage and a low value for dense vegetation. Another possible source for errors is from uncertainties in $C_{veg}$, which

reflects the thermal storage property of vegetated surface in the force-restore method. $C_{veg}$ was obtained through model calibration with UAS observed $T_s$. As shown in Figure 2, only three UAS data sets were available in the vegetated period. The insufficient model calibration may lead to uncertainties in $C_{veg}$.

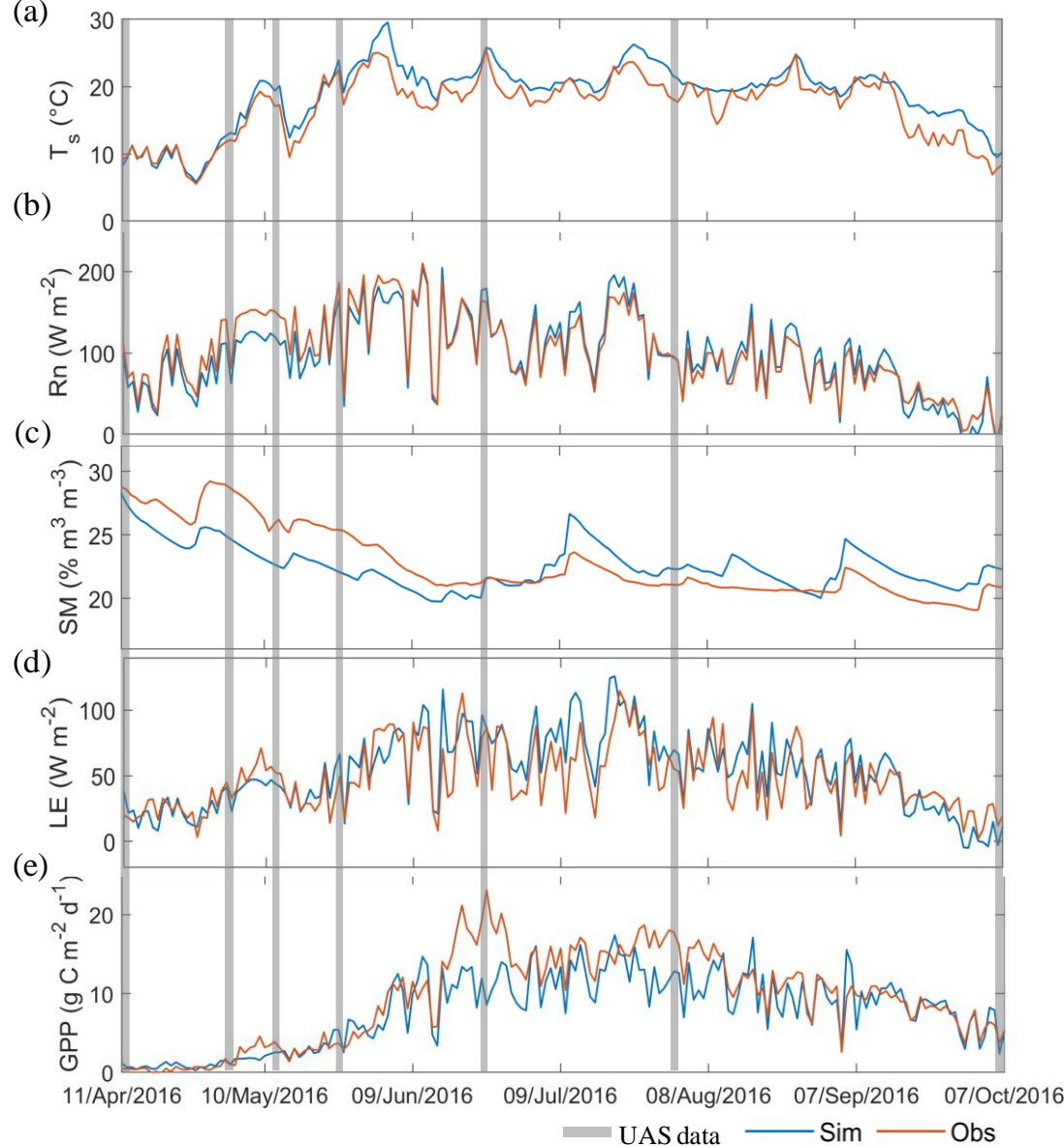

Figure 6. Simulated continuous daily land surface variables from 11th April to 7th October 2016 in the willow plantation. (a) Land surface temperature $T_s$, (b) Net radiation Rn, (c) Soil moisture $\theta$, (d) Latent heat flux LE, and (e) Gross primary productivity GPP. The dashed area indicates the time of acquired data for model calibration. The blue and red curves represent simulations and observations, respectively.

The estimated θ from SVEN achieved moderate performance in terms of errors and correlation. The model underestimates θ when NDVI is low, but overestimates θ with high NDVI as Figure 7 (c). Such errors may be due to the uncertainty in the model parameters related to θ. As shown in supplemental Table S5, the effective parameter values of Ks and n were taken as the mean values from the look-up table without considering ranges of variability (standard deviations in the table). In fact,

only one parameter, $SWS_{max}$, among the three parameters related to θ dynamics was calibrated with UAS estimates of θ in the root zone. To keep the model simple and parsimonious, the SVEN model only used one soil layer to simulate the dynamics of soil water storage (Figure 3). Similarly, the model also assumed that the residual soil moisture is equal to the soil wilting points. In the simulation of runoff generation, this simple model only considered the dominant runoff process, the "Dunne" mechanism (runoff occurs after soil water saturation, Dunne and Black, 1970) instead of the "Hortonian"

mechanism (runoff occurs when rainfall intensity exceeds the infiltration capacity, Horton, 1933), for this humid and flat site. Such model simplification could contribute to the relatively moderate performance of simulating θ. Additionally, UAS derived θ estimates used for calibration have errors of around 13% compared to the direct measurements (Wang et al., 2018a), which can induce uncertainties in the simulated time series through error propagates in the parameter calibration. Furthermore, only seven snapshot estimates from UAS were used to calibrate the model with an average frequency of 25

days during the period of fast growth. It can be expected that improving the UAS based estimates of θ, increasing the number of observations for model calibration, and adding more complexity of the model structure can raise simulation performance. For instance, when applying SVEN to other regions, the "Dunne" or "Hortonian" mechanism needs to be selected to simulate the surface water processes, according to the soil, vegetation and topographic conditions (Tauro et al., 2016).

The results of the simulated LE and GPP are shown in Figure 6 (d) and (e), respectively. In most cases, the simulation shows
the overestimation of LE, which closely relates to the estimates of Rn and θ. The simulation underestimated GPP, as the parameter $LUE_{max}$ was assumed to be the same as from a nearby beech forest (Wang et al., 2018a). Even though both sites are temperate deciduous forests, differences still exist between the natural beech forest and the willow forest bioenergy plantation. Notably, there is a significant underestimation of the simulated GPP in June of 2016 as shown in Figure 6 (e). Besides the possible uncertainties from the $LUE_{max}$ described above, the underestimation may also result from the

observation uncertainties in partitioning of GPP and respiration in eddy covariance data processing. In data processing, the night time net ecosystem exchanges were used to calculate the ecosystem respiration. During the night time, the eddy covariance footprint extended well-beyond the edges of the willow forest of interest, due to the stable atmospheric conditions. The tillage practices in the nearby Rapeseed fields (Fig. 1) could contribute overestimation of daytime ecosystem respiration, and thus leads to the overestimation of GPP in the eddy covariance data processing.

To check the model simulation performance under cloudy conditions, we analysed the relationship between model residuals and the ratio representing the diffuse radiation fraction (Figure 7 f-j). There were no significant differences for the residuals of the simulated $T_s$, Rn, θ, LE and GPP under low and high diffuse radiation fraction conditions. Due to the ability of UAS to acquire data in both cloud cover and clear sky conditions, the SVEN model was capable of interpolating land surface variables under cloud cover conditions with similar skill as under clear sky conditions.

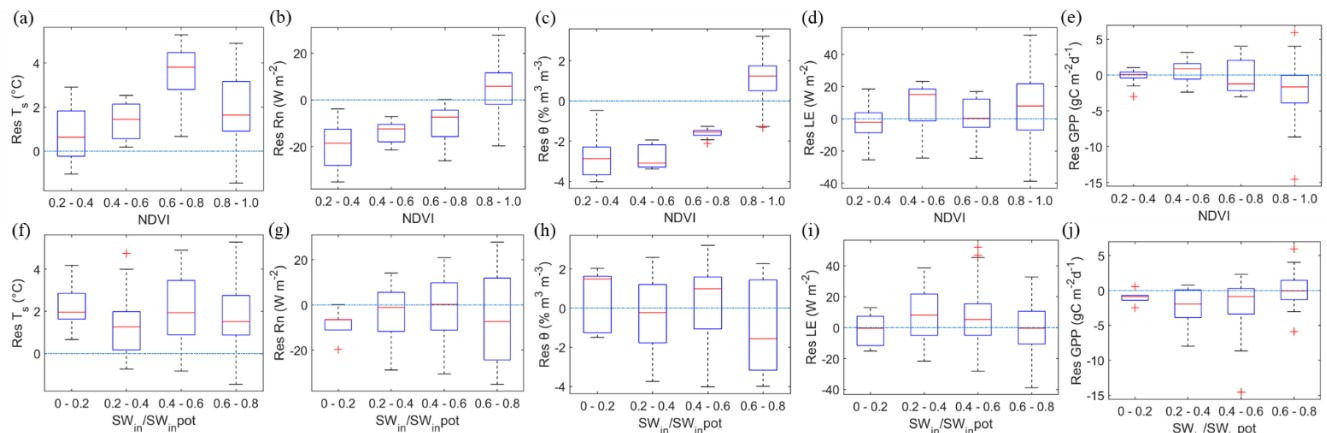

**Figure 7**. Boxplots of the residuals for the daily simulation. (a-e) are simulation residuals and NDVI. (f-j) are simulation residuals and the ratio of the actual ($SW_{in}$) and potential ($SW_{in}pot$) solar radiation, which is an indicator of the cloudiness condition. (a, f) surface temperature $T_s$, (b, g) net radiation Rn, (c, h) soil moisture $\theta$, (d, i) latent heat flux LE, and (e, j) gross primary productivity GPP. The blue dashed lines refer to the zero residuals.

## 4.3 Validation at half-hourly and monthly time scales

Validation of the half-hourly and monthly $T_s$, Rn, $\theta$, LE and GPP by the SVEN model is shown in Figure 8. The simulated half-hourly $T_s$, Rn, $\theta$, LE and GPP captured the temporal dynamics of land surface fluxes at this site. RMSDs for half-hourly $T_s$, Rn, $\theta$, LE and GPP are 3.04 °C, 63.82 W·m$^{-2}$, 1.99% m$^3$·m$^{-3}$, 56.37 W·m$^{-2}$ and 6.14 µmol·C·m$^{-2}$·s$^{-1}$, respectively. Compared to the simulation performance at the daily time scale (as shown in Table 3), the half-hourly simulation has higher RMSDs and lower $R^2$. Such performance may be due to that parts of SVEN modules are more suitable for daily scale simulation instead of the half-hourly. For instance, the simulation of LE in SVEN is based on the Priestley-Taylor equation originally applied to estimate monthly LE (Fisher et al., 2008) and was expended to be applied at daily steps (Garcia et al., 2013), but it is not appropriate for representing LE processes at sub-daily time scales.

Regarding the monthly time scale, RMSDs for $T_s$, Rn, $\theta$, LE and GPP are 2.10 °C, 10.96 W·m$^{-2}$, 1.86% m$^3$·m$^{-3}$, 9.09 W·m$^{-2}$ and 1.82 g·C·m$^{-2}$·d$^{-1}$, respectively. The monthly simulation has lower RMSDs and slightly higher $R^2$ compared to the daily simulation. The improvement of model performance from the half-hourly to daily and monthly time scales indicates the model errors can be reduced by aggregating the simulation outputs to longer time scales. Such accuracy also implies that the SVEN model has greater potential to temporally interpolate remote sensing observations at daily and monthly time scales, which are more relevant for applications in agriculture and ecosystem management.

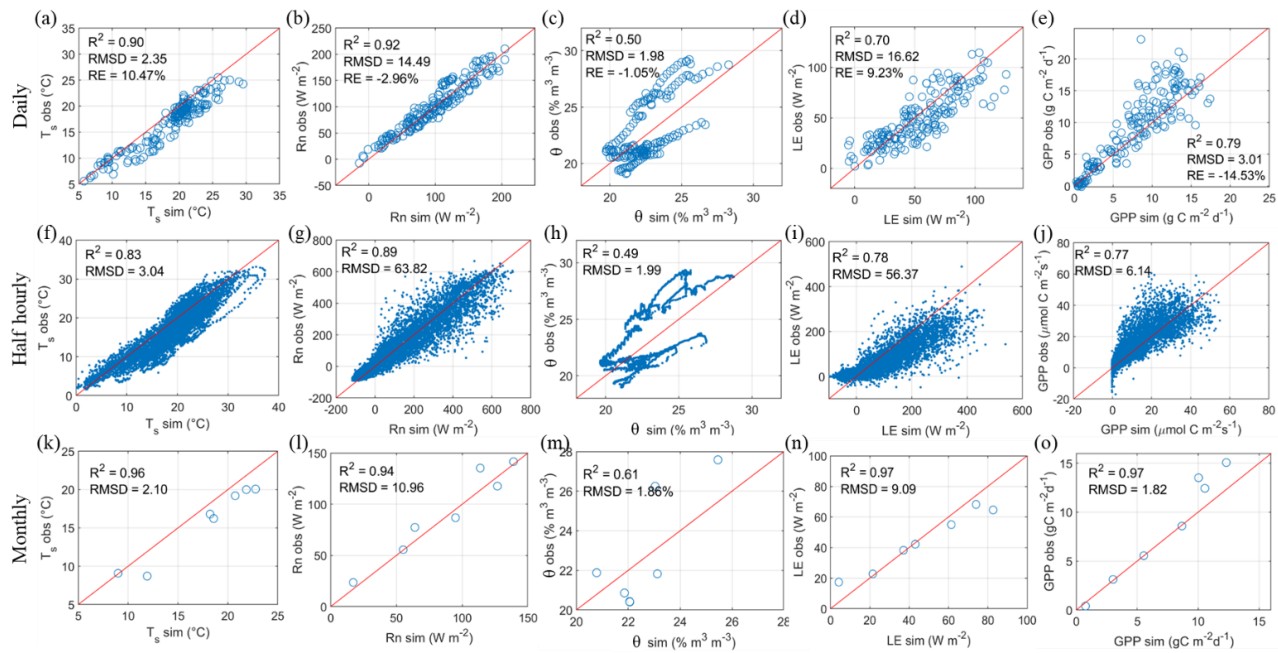

**Figure 8**. Validation of interpolated land surface variables at daily, half-hourly and monthly time scales in willow plantation. (a-e) are at daily scale, (f-j) are at half-hourly scale, and (k-o) are at the monthly scale. (a, f, k) surface temperature $T_s$, (b, g, l) net radiation Rn, (c, h, m) soil moisture $\theta$, (d, I, m) latent heat flux LE, and (e, j, o) gross primary productivity GPP. The metrics of RE for hourly and monthly scales are not shown, as they are the same as RE at the daily scale.

**Table 3**. Comparison of model simulation performance at half-hourly, daily and monthly time scales.

| Time scale | Statistics | Ts | Rn | $\theta$ | LE | GPP |
|---|---|---|---|---|---|---|
| | $R^2$ | 0.83 | 0.89 | 0.49 | 0.78 | 0.77 |
| Half hourly | RMSD | 3.04 °C | 63.82 W·m⁻² | 1.99% m³·m⁻³ | 56.37 W·m⁻² | 6.14 µmol·C·m⁻²·s⁻¹ |
| | NRMSD | 9.63% | 8.41% | 19.15% | 10.49% | 7.57% |
| | $R^2$ | 0.9 | 0.92 | 0.5 | 0.7 | 0.79 |
| Daily | RMSD | 2.35 °C | 14.49 W·m⁻² | 1.98% m³·m⁻³ | 16.62 W·m⁻² | 3.01 g·C·m⁻²·d⁻¹ |
| | NRMSD | 11.77% | 6.65% | 19.53% | 14.77% | 12.97% |
| | $R^2$ | 0.96 | 0.94 | 0.61 | 0.97 | 0.97 |
| Monthly | RMSD | 2.1 °C | 10.96 W·m⁻² | 1.86% m³·m⁻³ | 9.09 W·m⁻² | 1.82 g·C·m⁻²·d⁻¹ |
| | NRMSD | 18.49% | 9.29% | 25.91% | 17.88% | 12.42% |

## 4.4 Potential applications and improvement of SVEN

This study showed SVEN as a tool to temporally interpolate land surface variables between remote sensing acquisitions with few meteorological data. In statistical approaches, Alfieri et al. (2017) identified that a return interval of remote sensing observations should be no less than 5 days to accurately interpolate daily ET with relative errors less than 20%. The results

shown from our model based interpolation approach in the willow forest suggest that the revisit time for remote sensing observations can potentially be extended. For instance, seven instantaneous observations/simulations of this study with an averaged revisit time of 25 days can accurately interpolate the daily ET for 180 days. This comparison shows the benefits of using the model based approach to continuously estimate land surface fluxes from remote sensing based snapshot observations or estimates. The model based approach can be used to estimate ecosystem states and flux exchange with the atmosphere for a landscape (e.g. crop fields) with temporally sparse UAS flight campaigns. This approach has great potential for agricultural ecosystem monitoring and management. The interpolated continuous record of land surface variables can also further facilitate our understanding of the temporal dynamics of land surface-atmosphere flux exchanges.

On the other side, this study also provides ideas to utilize remote sensing estimates or observations to improve land surface modeling. Traditionally, the applicability of land surface models is limited due to complex model parameterization and the limited availability of "ground truth" or in-situ data for parameter calibration. As shown in this study, one solution for this limitation is using remote sensing based observations or estimates as "ground truth" for model calibration (Stisen et al., 2011; Zhang et al., 2009). This study calibrated the model parameters through remote sensing snapshot (UAS) estimates of land surface variables such as $T_s$ and $\theta$, and provided an example of integrating remote sensing data and process-based models. Other variables such as Rn, ET and GPP as shown in Figure 4 could also be incorporated for model calibration. Compared to complex land surface models, this approach is simple and efficient, especially suitable for operational applications to interpolate the remote sensing based snapshot estimates into the temporally continuous values.

Both the look-up table and parameter optimization approaches were used in this study to obtain the parameter values. For instance, we used a look-up table (Carsel and Parrish, 1988) to get values of n and $K_s$. The advantage of using the look-up table approach is that it can be easily applied according to the site conditions, such as vegetation types, soil texture and soil depth. However, this approach requires prior knowledge on the site. Insufficient knowledge of the site conditions may lead to the selection of unsuitable parameter values from the look-up tables. For instance, $K_s$ may vary at different soil layers and it could be difficult to select an effective $K_s$ to represent the condition for the entire soil layers. Regarding the optimization approach, this method has an advantage to achieve good fitting performance with UAS derived observations or estimates. However, this optimization approach needs to consider the number of observations and calibration parameters, parameter equifinality and multi-objective optimization (Her and Chaubey, 2015). For instance, due to limited fourteen UAS derived $T_s$ or $\theta$ available for calibration, we selected only four parameters ($C_{sat}$, b, $C_{veg}$, and $SWS_{max}$), which are hard to obtain from the look-up table approach with insufficient prior knowledge of the site, for optimization. To deal with parameter equifinality and multi-objective optimization, the Monte Carlo optimization was combined with the Pareto front analysis in this study. Other approaches e.g. Bayesian analysis could also be utilized to calibrate the model parameter with multiple objectives and separate the uncertainty sources: input, parameters and model structure (Vrugt et al., 2009). It can be a useful tool for the model calibration and quantification of the simulated uncertainty. Besides the look-up table and optimization approaches, another promising approach is to estimate soil or plant hydraulic properties from imaging spectroscopy (Goldshleger et al., 2012; Nocita et al., 2015) or thermal imaging data (Jones, 2004).

This model based interpolation approach can potentially also be applied with the space-borne remote sensing measurements to facilitate the temporally continuous estimation of large-scale land surface fluxes. The combination of the process-based models and satellite observations (e.g. Sentinel or MODIS land surface temperature and GPP products) can reduce the need of in-situ data for parameterizations. The temporally continuous estimates of land surface fluxes from satellite data facilitate our understanding of the temporal upscaling from instantaneous estimates to the daily or longer time scales to improve our knowledge of the coupled energy, water and carbon cycles at various temporal scales, particularly for data-scarcity regions. However, there are also challenges and limitations for widespread applications of the proposed model to other regions or with satellite Earth observation. SVEN also requires further improvement to enhance its ability for large-scale applications. For instance, the current soil moisture module in SVEN model is a simple water balance model with considering one soil layer, which has limited capacity to simulate soil water dynamics particularly in regions with complex landforms. In addition, the soil layer depth refers to the maximum root water uptake depth, which can vary with time (Guderle and Hildebrandt, 2015), but SVEN simplified this soil depth parameter to keep it consistent. Thus, in our study, SVEN only achieved moderate performance to simulate soil water dynamics and it can be expected that in water limited drylands, soil moisture simulation has a larger impact on the ET than in our site. Nonetheless, SVEN soil moisture estimates, relying on precipitation and water balance, should be in principle more accurate than those using thermal inertia (Garcia et al., 2013) , the original complementary approach relying on VPD (Fisher et al., 2008) or soil moisture proxies using antecedent precipitation proxies (Morillas et al., 2013; Zhang et al., 2010). Compared to the Penman-Monteith approach, the Priestley–Taylor approach may need adjustment of the aerodynamic term, when extending the study from radiation controlled sites to arid climates (Tadesse et al., 2018; Xiaoying and Erda, 2005). When applying SVEN to the large scale, the model needs to consider the sub-grid heterogeneity and identify the effective values for model parameters, e.g. soil saturated hydraulic conductance. The plant functional type and soil type parameterization scheme for different ecosystems and environmental conditions would be needed. Furthermore, there also remain challenges to get the reliability of atmospheric forcing such as radiation, precipitation and wind speed. Accurate gridded meteorological data from reanalysis, remote sensing or weather forecasting models as forcing will be needed. Moreover, satellite based observations or estimates may have larger uncertainties due to the coarser spatial resolution than UAS estimates. Applying SVEN with satellite data to large scale, we also need to evaluate the accuracy of satellite products and consider the error propagation from remote sensing estimates to the simulation outputs. In addition, satellite data in the optical and thermal ranges can only provide observations during cloudless conditions. Satellite data based model calibration may lead estimates biased toward sunny weather conditions.

## 5 Conclusion

Continuous estimation of land surface variables, such as surface temperature, net radiation, soil moisture, evapotranspiration and gross primary productivity at daily or monthly time scales is important for hydrological and ecological applications. However, remotely sensed observations were limited to directly estimate the instantaneous status of land surface variables at

the time of data acquisitions. Therefore, to continuously estimate land surface variables from remote sensing, this study developed a tool to fill the temporal gaps of land surface fluxes between data acquisitions and interpolate instantaneous estimates into continuous records. The tool is a dynamic Soil Vegetation Atmosphere Transfer model, the Soil-Vegetation, Energy, water and $CO_2$ traNsfer model (SVEN), which is a parsimonious model to continuously simulate land surface

variables with meteorological forcing and vegetation indices as model forcing. To interpolate the snapshot estimates from UAS, this study conducted the model parameter calibration to integrate the SVEN model and the snapshot estimates of surface temperature and soil moisture at the time of flight. Such model-data integration provides an effective way to continuously estimate land surface fluxes from remotely sensed observations. A case study was conducted with seven temporally sparse observations from UAS multispectral and thermal sensors in a Danish willow bioenergy plantation (DK-

RCW) during the growing season of 2016 (180 days). Satisfactory results were achieved with the root mean square deviations for the simulated daily land surface temperature, net radiation, soil moisture, latent heat flux and gross primary productivity equal to 2.35 °C, 14.49 $W \cdot m^{-2}$, 1.98% $m^3 \cdot m^{-3}$, 16.62 $W \cdot m^{-2}$ and 3.01 $g \cdot C \cdot m^{-2} \cdot d^{-1}$, respectively. This model based interpolation method has potential not just with UAS but also with remotely sensed data from other platforms, e.g. satellite and manned airborne systems, for a range of spatial and temporal scales.

*Data and code availability.* The data and code used in this study are available upon request from the corresponding authors.

*Supplement.* The supplement related to this article is available online.

*Author contribution.* All authors contributed to the design of this study and model development. MG and PBG contributed to funding acquisition. AI made contributions to the eddy covariance and meteorological data. SW conducted model

simulations and UAV data collection. SW wrote the original draft. All authors contributed to the discussion of results and the revision of this paper.

*Competing interests.* The authors declare that they have no conflict of interest.

*Acknowledgement:* The authors would like to thank the EU and Innovation Fund Denmark (IFD) for funding, in the frame of the collaborative international consortium AgWIT financed under the ERA-NET Co-fund Water Works 2015 Call. This

ERA-NET is an integral part of the 2016 Joint Activities developed by the Water Challenges for a Changing World Joint Programme Initiative (Water JPI). This study was also supported by the Smart UAV project from IFD [125-2013-5]. SW was financed from an internal PhD grant from the Department of Environmental Engineering at DTU and was financed by the COST action OPTIMISE for a short-term research stage. We would also like to thank the editor Dr. Nunzio Romano and three reviewers for the insightful suggestions and comments to improve the manuscript.

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
