# Peer review of "Temporal interpolation of land surface fluxes derived from remote sensing - results with an Unmanned Aerial System"

_Hydrology and Earth System Sciences, 2019_

## Referee Comment (RC1) · Anonymous Referee #1 · 26 Dec 2019

This study "Temporal interpolation of land surface fluxes derived from remote sensing results with an Unmanned Aerial System" developed a simple but operational land surface modeling framework, simulating energy balance, water and CO2 fluxes between the land surface and the. Unmanned aerial system (UAS) can be applied flexibly, and can have high spatial-temporal resolution data, which is used widely in recent decades. This study used UAS to provide optical and thermal data as model inputs for land surface-atmosphere fluxes monitoring. A dynamic soil vegetation atmosphere transfer model was developed here, together with the PT-JPL ET model and light use efficiency GPP model for simulating energy, water and CO2 cycles. The results showed that with using the data from UAS optical and thermal observations, the models were capable

to simulate the energy, water and CO2 fluxes in a deciduous tree plantation area, indicating that the UAS observations could be served as "ground truth" to calibrate soil and vegetation parameters, highlighting the usage of multiple remote sensing data for land-atmosphere flux monitoring. I think this manuscript is well written and the logic is pretty clear. The results are supported by the data shown here, while the authors explained the results adequately and clearly, though I have several minor questions on the current manuscript. (1) Introduction, why not introduce more about UAS? This is kind of a highlight of this study to use UAS data. Maybe include some introductions about recent studies using UAS data on GPP/ET simulations? (2) Why there is no UAS observation in July, and between May 25th and June 24th? In Fig. 2(c), the fIPAR seems to change a lot during 25/May to 24/June, thus, no observation during this time period may induce simulation errors in the model. (3) Why ignore the observation on 24/June when interpolate the UAS data. (4) Page 16, Ln. 2-3, not fully understand "This demonstrates that SVEN is capable to ....", syntax error? (5) Fig. 5(a), Ts, kind of systematic overestimation of Ts sim compared to Ts obs? So can the model parameters be calibrated to reduce the overestimation? (6) Fig 5(c), the scatterplot of SM sim and SM obs is kind of wired, which is more obvious in Fig. 7, I am wondering why? And also why not show daily results together with the half-hourly and monthly results in Fig. 7.
* * *

---

## Referee Comment (RC2) · Anonymous Referee #2 · 20 Jan 2020

This manuscript introduces a simple but effective coupled surface exchange model, with the goal to use it for gap filling of surface states and fluxes between measurements by remote sensing. The model requires higher resolution meteorological data as input for the forward simulation that serves as the gap filling procedure. The calibration is based on a very small number of snap shots of surface temperature and Normalized Difference Vegetation Index. As a proof of concept the method is applied using data obtained during seven flights of a drone, and continuous data from an eddy tower. The performance of the model es evaluated by comparing with independent eddy tower data of fluxes and states.

The manuscript presents an intriguing approach tested in a well designed study. The results are impressive, especially given the deliberate simplicity of the applied exchange model. The manuscript is well written manuscript. While I have some comments on the manuscript, I also recommend its publication in HESS and expect that it will find strong interest in the readership.

Major comments

(1) I found if very difficult to disentangle the different data sources used for the different application steps, which are: parameter estimation from literature and nearby observations calibration (UAS derived data, surface temperature and soil moisture) input for forward modeling (meteorological data from the eddy tower) validation of model output (independent eddy tower data) To make this more accessible I am missing an overview table systematically showing which data source was used for what purpose (as above). This would really help navigation,

(2) I would have liked to see some more discussion on the next challenges for the more widespread application of the proposed method with less ideal input data for the forward model. What are the expected limitatations of the approach? Currently the discussion regarding this point is very short. For example, the discussion mentions that the method could be extended to larger scales by using online weather data. However, those have also higher uncertainty compared to the data from the tower. Also, the JPL-Priestley-Taylor-ET estimate is less reliable in more arid climates which probably requires additional adjustments in those conditions, etc. I recommend enhancing the discussion regarding this.

(3) I am confused about what is the underlying hypothesis motivating the comparison of the residuals across different stages of diffuse light conditions? The analysis is motivated by stating that remote sensing is typically biased towards collection in direct sunlight conditions. But this was probably not the case in your exercise, since you were collecting data from a drone. Therefore the calibration data set should not be affected

by this bias? Why are you expecting the bias in the residuals?

(4) I find the equations of the manuscript difficult to read because the abbreviations of the variables are of several letters. I understand that in some instances this is done to adhere by the nomenclature in the discipline, e.g converting LAI to a one letter variable would probably cause confusion. But in most cases this is not an issue. For example, radiation can be abbreviated with R and the components by indices, fluxes with Q or J with indices. Also canopy storage, soil water storage etc. This would also increase consistency. I strongly recommend incorporating the one letter abbreviation paradigm as much as possible. See also HESS author guidelines (Mathematical requirements) https://www.hydrology-and-earth-system-sciences.net/for_authors/manuscript_preparation.html

Detailed comments

Abstract, Page 1

Line 18: "SVEN interpolated the snapshot Ts, Rn, SM, ET and GPP to continuous records" This phrase is confusing, as it sounds like measurements of each of those variables were used, when according to the methods section only Ts and NDVI were used for calibration.

Line 21-22 I would not mind, if the errors were not stated quantitatively here, but if this is desired: An indication of the errors in percent would be more meaningful.

Introduction

Line 19/20: I think you mean "high persistence"

Methods

Page 9, Line 5 "low pass filter for $T\_s$": Can you be more specific about the cutoff frequency? Which interval does this roughly refer to?

Page 9 Line 24 Wind speed seems to be one of the variables that need to be available

continuously to apply the method. Is it reasonable to have such good knowledge of the wind speed? How sensitive is it?

Page 10, Line 15-20 The PF-JPL works much better in temperate then drier climate. Your appraisal does not mention this limitation, but I think it may be important for applying this method more generally. Could you add a note on this, either here or in the discussion?

Page 11, Line 27 should probably be "equation 29" instead of "equation 28"

Page 12 Line 2 Soil water storage has different units here (m) and on page 9, Line 10 (mˆ3). I think it is fine to stick with m.

Page 12, Eq. 30-32, Page 13 Line 19-20 I am not sure how theta_r and theta_s are dealt with? They are not calibrated and not mentioned for the look-up table. Based on Table S5, where they are included, I am assuming they were looked up too. But please be more specific and include them in the list of parameters in Table 2.

Page 13 Table 2 It will help navigating the text, if in the table included a column indication of whether this parameter was looked up or calibrated in this study. I suggest adding this.

Page 13 Line 22 In my understanding calibrating SWS_max boils down to calibration the root water uptake depth?If yes, would be good to indicate this. While I have no objections against this procedure here, I conjecture that root water uptake depth may vary with time over the growing season. Thus, this may be a limitation of the model, which could be mentioned in the discussion.

Page 13, Line 7-9, Supplement Table S3 Please add the values for each of the initial conditions.

Results

Page 15, Section 4.1 Not sure whether I overlooked this, but can you please indicate

the values of the calibrated parameters? Also: I like Fig 4 showing the objective function. Near the pareto optimum plot a number of potentially very good model runs. Are they all roughly similar parameter values or do they differ substantially? This would give an indication of how well defined this model is in terms of the processes that are represented or/and the sensitivity of some of the parameters. Can you comment on this?

Page 15 Lines 19-20, Page 18, Lines 16-20. I feel the numbers are crowding the text, and are difficult to take in. It is enough to refer to Fig 5, Fig 7 or alternatively collect them in a Table.

Page 16, Line 5, Line 8 To me Ts does not appear to be underestimated only in high NDVI conditions. Ts is also underestimated in May, when GPP is still very low. I am not convinced of this distinction .. but in order to support your point, you could color the points in the top right panel of Fig 5 with shades indicating NDVI (or GPP).

Page 16, Line 24, Fig 5 Would be good to indicate the times of the seven snapshots in Fig 5 by vertical lines (solid for all UAS, dashed for augmented with tower data), so it is easier to see when the data was obtained for calibration.

Page 16, Line 28 Do you mean "nearby" instead of "nearly"?

Page 18, Line 1 I think "be" should be erased

Page 18, Line 4-5, Fig 6 Can you please indicate in Fig 6 what the red lines refer to? I am at a loss, especially in panel (a). Also, I am not sure how the conclusion "GPP was underestimated under diffuse radiation conditions" is seen from the Figure, I am assuming in panel (j). Does the point cloud show a trend?

Page 18, Line 6 Add "of" after enhancement

Fig 7 Fonts in the top and bottom panels are not the same. Fig 5 & Fig 7 I was confused at first about the difference of the Fig 7 to the right panels in Fig 5. I concluded they are the same, just showing different time intervals. Can you collect them in one Fig? It

[Figure]

would be easier to compare.

---

## Referee Comment (RC3) · John Kochendorfer (Referee) · 3 Feb 2020

**General comments**

"Temporal interpolation of land surface fluxes derived from remote sensing – results with an Unmanned Aerial System" describes the use of a suite of simple models to interpolate surface fluxes and surface state variables between sporadically available land surface measurements. A model dubbed the SVEN was created by augmenting a Priestly Taylor model with new components to enable its use at timescales as short as 30-min. Instantaneous remotely sensed variables recorded mainly using a UAS were used to calibrate the model, and the model was then used to fill in the extensive gaps between measurements. This work introduces and demonstrates of the technique, which was designed to be used with both satellite and UAS remote sensing measurements. It is a solid manuscript, with room for some improvement.

10  Because one of the primary stated goals of the paper is the development of an application to satellite remote sensing, the omission of actual satellite measurements is conspicuous. I suggest that more attention be given to the topic of using satellite data. For example, what might be the shortcomings of applying the model to satellite-based measurements? Were UAS measurements relied upon for this paper rather than space-based measurements due to the inadequate spatial resolution of satellite measurements?  At this site in-situ and UAS measurements are

15  available (and used), but how well will the model work for the rest of the Earth's surface? Clarify the purpose of the model (including the parameter fitting) in the broadest sense, and develop, test, and describe the results accordingly.

In addition, more care needs to be taken with the way remotely sensed measurements are handled. They are misleadingly referred to as "ground truth" or direct measurements throughout the manuscript, when most of the

20  variables derived from remote sensing data were modeled or inferred, rather than measured directly. Uncertainties due to this also require more attention.

 The writing should be reviewed carefully by a native English speaker. Some examples are included in the specific comments below, but the manuscript includes many errors in writing and sentence structure.

**Specific comments**

25  P 1, ln 7-8. With the exception of $T_s$, all of these variables are determined using remote sensing products based on a suite of different models and assumptions. For example, different vegetation indices can be measured remotely, but GPP cannot. The same applies to ET, SM, and Rn – none of these variables are measured directly using remote sensing, but the first sentence misleadingly indicates otherwise. Without detracting from main point of this sentence, a word such as "inferred" or "derived" could easily be included for more accuracy.

30  P 1, ln 20. Delete the word, "well" from, "…SVEN can *well* estimate…". Awkward as written.

P 2, ln 2-3. "Minimum parameterization" is awkward as written.

P 2, ln 7. "Mostly needed" is awkward as written. Also, replace "high frequency" with "prevalence."

P 2, ln 11. Replace "flexibly" and "favorable" with more precise descriptors.

P 2, ln 14. Replace "still *just* provide" with "still *only* provide."

35  P 2, ln 16. Replace "uncovered" with "unknown."

P 2, ln 25. "using statistical interpolation could be challenging…" is awkward as written.

P 2, ln 28-29. "can be better" is awkward as written.

P 2, ln 30. Delete "a" in, "in a variable climate conditions."

P 3, ln 6. "as for example the turbulent fluxes are typically…" is awkward as written.

P 3, ln 7. "simpler but operational models based interpolation" is awkward as written.

5   P 3, ln 11. Delete "the" at the beginning of this line.

P 3, ln 15. Rewrite as, "limited meteorological inputs, and parameters that…".

P3, ln 21-22. "now becomes" is awkward as written.

P4, ln 4. Add the word, "it" after "forcing".

P4, ln 17. Change, "onboard have been conducted" to, "onboard were conducted." And "Details refer…" to, "For
10  more details refer…".

Figure 1. This is exactly the same as Figure 1 from Wang et al. (2018b). I don't know HESS's rules regarding this type
of thing, so I will refer to the Editor for guidance. I would never reuse a figure like this myself, but if this is actually
acceptable, the original usage should certainly be referenced. In addition, a small wind rose would be a nice
addition to the figure; at a measurement height of 10 m (Wang et al., 2018b), the flux footprint will extend well
15  beyond the edges of the figure in some conditions. As an aside to be passed onto the site manager, if the eddy
covariance instrumentation were closer to the top of the canopy, it would help alleviate this problem."

P5, ln 24. "Data of few UAS flight campaigns" is awkward as written.

P5, ln 26. Replace the word, "resemble" with more appropriate verbiage.

P5, ln 27. Clarify that the "ground truth" SM measurements were not actual SM measurements, and describe the
20  uncertainty and shortcomings of the remotely sensed SM product in detail.

P5, ln 29. "which corresponded to the willow emerging period with a high growth rate" is awkward as written.

P7, ln3. "and can facilitates to temporally interpolate" is awkward as written.

P8, ln 24-26. Clarify that this includes the existence of a canopy. As written, it reads like a simple soil diffusion-
based approach, that neglects the existence of vegetation. There is a transfer coefficient for the canopy (Cveg)
25  described on P9, along with LE etc., so I assume this all adds up correctly (I am not a modeler), but a more complete
initial description is wanted on P8.

P9, ln 24-25. Change to, "k is *the* von Karman constant."

P12, ln 20. "The rest of constraints," is awkward as written.

P12, ln 21. "are the same modifying," is awkward as written.

30  P13, ln 6. Change to, "UAS-*derived* observations," or otherwise clarify that many of these UAS variables were not
measured directly.

P14, ln 1. Change, "facilitate," to, "facilitates."

Eq 34 description. Clarify what time period (e.g. 30 min or 24 h) was used for this EC measurement adjustment, and how missing data were handled.

P14, ln 25. "well represent" is awkward as written.

Validation at the daily time scale Section. Augment the discussion of uncertainty in the UAS-derived measurements (as compared to direct measurements).

P16, ln 24. "that the better UAS based snapshot estimates of SM…" is awkward as written. Perhaps, "that improving the UAS-based estimates of SM…".

P16, ln 34. "has a large coverage" is awkward as written. More accurately, it could be replaced with something like: "extended well-beyond the edges of the Willow forest of interest".

P18, ln 1. "are be good" is awkward as written.

P18, ln 3. "do not show difference" is awkward as written.

P18, ln 5. "do to that the model" is awkward as written.

P18, ln 6. "enhancement diffuse radiation effects" is awkward as written.

P18, ln 19. Perhaps change, "R2 for Ts…" to, "R2 for *monthly* Ts…"

P20, ln 10. Change, "understanding on the" to, "understand of the".

Conclusions. What would the effects of using space-based remote sensing measurements be, rather that UAS measurements? Also discuss how well this method will work in areas where in-situ measurements are unavailable to better parameterize the UAS and SVEN measurements.

Equation and variable abbreviations. I cheated and read the other Reviewer Comments. I disagree with Referee #2 regarding their objection to the use of multiple letter abbreviations; I am already familiar with LUE, PAR, GPP, ET, etc., so their usage made it easier for me to follow the manuscript. In addition (this may have more to with my background than what is most suitable for HESS), I am more accustomed to Ɵ than SM for soil moisture, and R (surface runoff) could easily be confused for respiration (although honestly I am not sure if there is a more widely used abbreviation for runoff).

---

## Author Comment (AC1) · 30 Mar 2020

Response to the review of Referee 1. We have copied the comments of the referee hereunder with our comments appearing after the referee's comments.

This study "Temporal interpolation of land surface fluxes derived from remote sensing results with an Unmanned Aerial System" developed a simple but operational land surface modeling framework, simulating energy balance, water and $CO_2$ fluxes between the land surface and the. Unmanned aerial system (UAS) can be applied flexibly, and can have high spatial-temporal resolution data, which is used widely in recent decades. This study used UAS to provide optical and thermal data as model inputs for land

surface-atmosphere fluxes monitoring. A dynamic soil vegetation atmosphere transfer model was developed here, together with the PT-JPL ET model and light use efficiency GPP model for simulating energy, water and CO2 cycles. The results showed that with using the data from UAS optical and thermal observations, the models were capable to simulate the energy, water and CO2 fluxes in a deciduous tree plantation area, indicating that the UAS observations could be served as "ground truth" to calibrate soil and vegetation parameters, highlighting the usage of multiple remote sensing data for land-atmosphere flux monitoring. I think this manuscript is well written and the logic is pretty clear. The results are supported by the data shown here, while the authors explained the results adequately and clearly, though I have several minor questions on the current manuscript.

Reply: Thank you for the insightful comments and suggestions, which are very helpful to improve the manuscript. We totally agree that the great potential of utilizing UAS for monitoring land surface energy, water and CO2 processes. The proposed model in our study is capable of temporal interpolating the remote sensing based snapshot estimates into the continuous records. Here, we have addressed your comments point-by-point.

(1) Introduction, why not introduce more about UAS? This is kind of a highlight of this study to use UAS data. Maybe include some introductions about recent studies using UAS data on GPP/ET simulations?

Reply: Thank you for the comments and suggestions. We have revised the introduction to add more review contents about UAS, particularly on applying UAS data for GPP / ET estimation. Please see Line 13-20 on P2 in the revised clean version.

(2) Why there is no UAS observation in July, and between May 25th and June 24th? In Fig. 2(c), the fIPAR seems to change a lot during 25/May to 24/June, thus, no observation during this time period may induce simulation errors in the model.

Reply: Thank you for the comments and suggestions. We totally agree with the reviewer's opinion on the importance of collecting observations during the period from May 25th to June 24th. However, due to technical issues, we did not manage to fly UAS over that period. On the other side, this low frequency of collecting UAS observations provides an opportunity to demonstrate that the "ground truth" collected from sparse remote sensing observations can be utilized to be temporally interpolated to obtain the continuous estimates.

(3) Why ignore the observation on 24/June when interpolate the UAS data.

Reply: Thank you for the comments and suggestions. We do incorporate the observation on June 24th into the temporal interpolation, but the observation on June 24th is not from UAS. The observations on that day are from the ground PAR sensors (Table 1). Due to technical issues, we did not manage to fly UAS over that period. However, to demonstrate the potential to use the proposed SVEN model to temporally interpolate the snapshot estimates, we have incorporated the ground IPAR observations on June 24th to simulate the process of vegetation growth in this period. To make the context clearer, we have revised the sentence on L10-15 on P6.

(4) Page 16, Ln. 2-3, not fully understand "This demonstrates that SVEN is capable to : : :.", syntax error?

Reply: Thank you for the comments and suggestions. We have revised this sentence. It should be that "Such simulation accuracy demonstrates that SVEN is capable of temporal interpolating the snapshot estimates or observations between remote sensing acquisitions to form continuous daily records." Please see L11-13 on P17.

(5) Fig. 5(a), Ts, kind of systematic overestimation of Ts sim compared to Ts obs? So can the model parameters be calibrated to reduce the overestimation?

Reply: Thank you for the comments and suggestions. Yes, we can try to reduce the systematic overestimation of Ts through calibration. However, this study used multi-objective calibration procedures to consider both Ts and soil moisture. As results

shown in the Pareto Front of Figure 5, if we want to obtain better performance of simulating Ts, the performance of simulated soil moisture could be degraded. Thus, based on the Pareto front in Figure 4, we choose the parameter sets to achieve relatively good simulations for both Ts and soil moisture. To make this context clearer, we have revised the manuscript. Please find the revised sentences of L20-25 on P16.

(6) Fig 5(c), the scatterplot of SM sim and SM obs is kind of wired, which is more obvious in Fig. 7, I am wondering why? And also why not show daily results together with the half-hourly and monthly results in Fig. 7.

Reply: Thank you for the comments and suggestions. They are very helpful. There are several reasons for the moderate performance of simulating soil moisture in this study. Such model performance may be due to the uncertainty in the model parameters related to $\theta$. As shown in supplemental Table S5, the effective parameter values of the infiltration rate for the saturated soil (Ks) and fitting parameter of the Mualem model (n) were taken as the mean values from the look-up table without considering ranges of variability (standard deviations in the table). In fact, only one parameter, SWSmax, among the three parameters related to $\theta$ dynamics was calibrated with UAS estimates of $\theta$ in the root zone. To keep the model simple and operational, the SVEN model only used one soil layer to simulate the dynamics of soil water storage (Figure 3). Such simplification could also contribute to the relatively moderate performance of simulating $\theta$. Additionally, UAS derived $\theta$ estimates used for calibration have errors of around 13% (Wang et al., 2018a), which can induce uncertainties in the simulated time series through error propagates in the parameter calibration. Furthermore, only seven snapshot estimates from UAS were used to calibrate the model with an average frequency of 25 days during the period of fast growth. It can be expected that improving the UAS based estimates of $\theta$ and increasing the number of observations for model calibration can improve the simulation performance. To elaborate details on the simulation performance of soil moisture, we have added discussion in L3-13 on P19. Thank you for the suggestion on the figure. We agree that combining daily results together with the halfhourly and monthly results could be better. We have revised Figure 7 and combined it with Figure 5 according to the reviewer's suggestions.

---

## Author Comment (AC2) · 30 Mar 2020

Response to the review of Referee 2. We have copied the comments of the referee hereunder with our comments appearing after the referee's comments.

1. This manuscript introduces a simple but effective coupled surface exchange model, with the goal to use it for gap filling of surface states and fluxes between measurements by remote sensing. The model requires higher resolution meteorological data as input for the forward simulation that serves as the gap filling procedure. The calibration is based on a very small number of snap shots of surface temperature and Normalized Difference Vegetation Index. As a proof of concept the method is applied using data

obtained during seven flights of a drone, and continuous data from an eddy tower. The performance of the model es evaluated by comparing with independent eddy tower data of fluxes and states. The manuscript presents an intriguing approach tested in a well designed study. The results are impressive, especially given the deliberate simplicity of the applied exchange model. The manuscript is well written manuscript. While I have some comments on the manuscript, I also recommend its publication in HESS and expect that it will find strong interest in the readership.

Reply: We appreciate the reviewer's insightful comments and suggestions, which were very helpful to improve the manuscript. We totally agree that the great potential of utilizing the simple but effective land surface models to fill gaps between observed surface states and fluxes from remote sensing. Here, we have addressed your comments point-by-point.

Major comments 2. I found if very difficult to disentangle the different data sources used for the different application steps, which are: parameter estimation from literature and nearby observations calibration (UAS derived data, surface temperature and soil moisture) input for forward modeling (meteorological data from the eddy tower) validation of model output (independent eddy tower data) To make this more accessible I am missing an overview table systematically showing which data source was used for what purpose (as above). This would really help navigation, Reply: Thank you for your suggestions. To make the data and parameter sources clear, we have added one figure on the flow chart of this study. Please see Figure 4, which includes details on the model inputs, parameter, outputs and calibration procedures. We also have added the sources of parameter values into the figure. For details, please refer to L1-5 on P14 in the revised clean version. I would have liked to see some more discussion on the next challenges for the more widespread application of the proposed method with less ideal input data for the forward model. What are the expected limitations of the approach? Currently the discussion regarding this point is very short. For example, the discussion mentions that the method could be extended to larger scales by using

online weather data. However, those have also higher uncertainty compared to the data from the tower. Also, the JPL-Priestley-Taylor-ET estimate is less reliable in more arid climates which probably requires additional adjustments in those conditions, etc. I recommend enhancing the discussion regarding this.

Reply: Thank you for your suggestions. We agree that there are still challenges and limitations for the more widespread application of the proposed model, particularly when applying models to the large scales and data-scarcity regions. First of all, the SVEN model is a very simple and parsimonious process-based model. For instance, the current soil moisture module in the SVEN model is a simple water balance model with considering one soil layer, which has limited capacity to simulate soil water dynamics particularly in regions with complex landforms. In addition, the soil layer depth refers to the maximum root water uptake depth, which can vary with time, but SVEN simplified this soil depth parameter to keep it consistent. Thus, in our study, SVEN only achieved moderate performance to simulate soil water dynamics and it can be expected that in water limited drylands, soil moisture simulation has a larger impact on the ET than in our site. Additionally, compared to the Penman-Monteith approach, the Priestley–Taylor approach may need adjustment of the aerodynamic term, when extending the study from radiation controlled sites to arid climates. Regarding the model-data integration, our study used a two-objective optimization scheme, there are more advanced algorithms e.g. data assimilation could enable the consideration of data and model uncertainties in the integration process. Moreover, when applying the model with satellite coarse resolution data to the large scale, there will be four major impacts. First, the space-borne remote sensing data have much coarser spatial resolution. If we move the simulation to the large scale with satellite data, we need to find accurate gridded meteorological data as forcing. UAS imagery has limited coverage and thus this study only used one meteorological station data as forcing. As satellite data have coarser pixel sizes, we also need to consider the sub-grid heterogeneity and identify the effective values for model parameters. Note all parameter values of models were obtained from parameter calibration with remote sensing based estimates. For instance, in our study,

we used the look-up tables with soil texture information to identify soil parameter values. In the large-scale simulation with satellite data, the plant functional type and soil type parameterization scheme for different ecosystems and environmental conditions would be needed. However, the integration of accurate remote sensing estimates with land surface models would be beneficial to reduce the dependency of plant functional type parameterization schemes and achieve a higher accuracy to predict land surface variables. In addition, coarse resolution satellite data may have limited accuracy to predict land surface fluxes compared to the detailed UAS data. Applying SVEN with satellite data to large scale, we also need to be careful about the accuracy of remote sensing based estimates and the error propagation from the model inputs to the outputs. Satellite data in the optical and thermal ranges can only provide observations during the sunny weather conditions. However, the UAS data in this study were collected in both sunny and cloudy conditions. We envision that using satellite based data to calibrate model may lead the model estimates biased towards the sunny conditions. We also agree with the reviewer that compared to the Penman-Monteith approach, the Priestley–Taylor approach may need adjustment of the aerodynamic term, when extending the study from radiation controlled sites to arid climates. We have added these contents regarding the model improvement and challenges to the discussion part. For details, please refer to section 4.4 in the revised clean version.

3. I am confused about what is the underlying hypothesis motivating the comparison of the residuals across different stages of diffuse light conditions? The analysis is motivated by stating that remote sensing is typically biased towards collection in direct sunlight conditions. But this was probably not the case in your exercise, since you were collecting data from a drone. Therefore the calibration data set should not be affected by this bias? Why are you expecting the bias in the residuals?

Reply: Thank you for your comments. We have revised Figure 7 to be boxplots to make the results clear. We agree that due to that UAS data collection happens on both sunny and cloudy weather conditions, we did not see significant differences of

residuals in simulating surface temperature, net radiation, soil moisture, latent heat flux, and gross primary production for different sky conditions. We have revised the description and for details, please refer to L6-10 on P20.

4. I find the equations of the manuscript difficult to read because the abbreviations of the variables are of several letters. I understand that in some instances this is done to adhere by the nomenclature in the discipline, e.g converting LAI to a one letter variable would probably cause confusion. But in most cases this is not an issue. For example, radiation can be abbreviated with R and the components by indices, fluxes with Q or J with indices. Also canopy storage, soil water storage etc. This would also increase consistency. I strongly recommend incorporating the one letter abbreviation paradigm as much as possible. See also HESS author guidelines (Mathematical requirements) https://www.hydrology-and-earth-systemsciences.net/for_authors/manuscript_preparation.html

Reply: Thank you for your suggestions. We have revised the abbreviations of variables to be one letter abbreviation as much as possible. For instance, we used ALB to represent surface albedo in the previous version. In the revised version, we used one letter abbreviation A to stand for surface albedo. Please see L15 on P8. (Notably, most studies used the Greek letter $\alpha$ to represent surface albedo. However, $\alpha$ has already been used as the PT coefficient in Eq. 22.) We also have changed soil moisture (SM) to one Greek letter $\theta$. We have changed the wind speed from WS to u. Furthermore, we have also summarized all abbreviations in the supplementary material.

Detailed comments 5. Abstract, Page 1 Line 18: "SVEN interpolated the snapshot Ts, Rn, SM, ET and GPP to continuous records" This phrase is confusing, as it sounds like measurements of each of those variables were used, when according to the methods section only Ts and NDVI were used for calibration.

Reply: Thank you for your suggestions. We have revised this sentence to be clearer. Based on model parameter calibration with the snapshots of land surface variables at

the time of flight, SVEN interpolated the UAS based snapshots to continuous records of Ts, Rn, $\theta$, ET and GPP for the growing season of 2016 with forcing from continuous climatic data and NDVI. Please see L17-19 on P1.

6. Line 21-22 I would not mind, if the errors were not stated quantitatively here, but if this is desired: An indication of the errors in percent would be more meaningful.

Reply: Thank you for your suggestions. In order to make clearer, we have added the statistics to be in percent (normalized root-mean squares deviations, NRMSD). The NRMSD was calculated as the ratio between root-mean squares deviations and the range (maximum minus minimum) of observations. Please see L20-21 on P1.

7. Introduction Line 19/20: I think you mean "high persistence"

Reply: Thank you for your suggestions. It is a mistake. We have changed the word to "high persistence". Please see L25 on P2.

8. Methods Page 9, Line 5 "low pass filter for T_s": Can you be more specific about the cutoff frequency? Which interval does this roughly refer to?

Reply: Thank you for your suggestions. The cutoff frequency is 24 hours. We have revised the sentence in L21 on P9. T_d refers to the deep soil temperature ($^\circ$C) calculated by applying a low-pass filter to T_s with the cut-off frequency of 24 hours.

9. Page 9 Line 24 Wind speed seems to be one of the variables that need to be available continuously to apply the method. Is it reasonable to have such good knowledge of the wind speed? How sensitive is it?

Reply: Thank you for your suggestion. Yes, the model needs the wind speed as inputs to calculate the aerodynamic resistance for estimating sensible heat fluxes. The accurate information about the wind speed is important for the model to estimate the aerodynamic resistance to the transfer of sensible heat flux. Wind speed, however, it is not used to estimate the transfer of vapor flux (evapotranspiration) as we used a Priestley-Taylor JPL equation. The PT-JPL model used the PT coefficient ($\alpha$) with a

fixed value to account for the ratio between aerodynamic term and radiation. Thus, the ET is not sensitive to wind speed in the model. The larger contribution to errors in H is actually from the soil, canopy, and air temperature (Chehbouni et al., 2001). After that, uncertainties in soil and canopy emissivity values, canopy height, and wind speed also have measurable effects on the accuracy of simulating H (Sánchez et al., 2008). In addition, the error in the sonic anemometer is very low. With traditional cup anemometers, a larger error, of about 10% of error in the wind speed will translate in an error in H of about 5-10% (depending on the temperature difference) for the type of vegetation in this paper. In SVEN the surface temperature estimates depend on the energy forcing which is constrained by three different energy variables (Rn, H, LE) and soil moisture, apart from the temperature from the previous time step. Therefore, errors in wind speed only affect H should not affect too much the temperature estimates. However, we also agree that without field measurements such as the sonic anemometer, the wind speed data could have large uncertainties from weather forecasting or climate reanalysis data. Applying the SVEN model to the large scale or other data-scarcity regions could have more uncertainties from wind speed data. Thus, we have added these discussions about the uncertainties from wind speed to model performance. Please see L28-29 on P23. Chehbouni, A., Nouvellon, Y., Lhomme, J. P., Watts, C., Boulet, G., Kerr, Y. H., ... & Goodrich, D. C. (2001). Estimation of surface sensible heat flux using dual angle observations of radiative surface temperature. Agricultural and Forest Meteorology, 108(1), 55-65. Sánchez, J. M., Kustas, W. P., Caselles, V., & Anderson, M. C. (2008). Modelling surface energy fluxes over maize using a two-source patch model and radiometric soil and canopy temperature observations. Remote sensing of Environment, 112(3), 1130-1143.

10. Page 10, Line 15-20 The PF-JPL works much better in temperate then drier climate. Your appraisal does not mention this limitation, but I think it may be important for applying this method more generally. Could you add a note on this, either here or in the discussion?

Reply: Thank you for your suggestion. We agree that PT-JPL works better in temperate than drier climate. We also agree that it is good to mention the limitation of this model. We have added this suggestion to L24-26 on P23. Compared to the Penman-Monteith approach, the Priestley–Taylor approach may need adjustment of the aerodynamic term, when extending the study from radiation controlled sites to arid climates (Tadesse et al., 2018; Xiaoying and Erda, 2005). Tadesse, H. K., Moriasi, D. N., Gowda, P. H., Marek, G., Steiner, J. L., Brauer, D., Talebizadeh, M., Nelson, A. and Starks, P.: Evaluating evapotranspiration estimation methods in APEX model for dryland cropping systems in a semi-arid region, Agric. Water Manag., doi:10.1016/j.agwat.2018.04.007, 2018. Xiaoying, L. and Erda, L.: Performance of the Priestley-Taylor equation in the semiarid climate of North China, Agric. Water Manag., doi:10.1016/j.agwat.2004.07.007, 2005.

11. Page 11, Line 27 should probably be "equation 29" instead of "equation 28" Reply: Thank you for your suggestion. We have revised it.

12. Page 12 Line 2 Soil water storage has different units here (m) and on page 9, Line 10 (mĚĘ3). I think it is fine to stick with m. Reply: Thank you for your suggestion. We have revised all units for soil and canopy water storage to be m.

13. Page 12, Eq. 30-32, Page 13 Line 19-20 I am not sure how theta_r and theta_s are dealt with? They are not calibrated and not mentioned for the look-up table. Based on Table S5, where they are included, I am assuming they were looked up too. But please be more specific and include them in the list of parameters in Table 2.

Reply: Thank you for your suggestion. theta_r and theta_s are from the look-up tables based on soil texture. I have revised Table 2 to include theta_r and theta_s. For details, please refer to Table 2 and Figure 4.

14. Page 13 Table 2 It will help navigating the text, if in the table included a column indication of whether this parameter was looked up or calibrated in this study. I suggest adding this.

Reply: Thank you for your suggestion. We have revised Table 2 and added one column to indicate the source of parameter values (model calibration or look-up table). Furthermore, we have added Figure 4 to show the model implementation of this study.

15. Page 13 Line 22 In my understanding calibrating SWS_max boils down to calibration the root water uptake depth?If yes, would be good to indicate this. While I have no objections against this procedure here, I conjecture that root water uptake depth may vary with time over the growing season. Thus, this may be a limitation of the model, which could be mentioned in the discussion.

Reply: Thank you for your suggestion. We agree that the root water uptake depth vary with time over the growing season. Our paper aims to propose a simple but operational model for interpolation of land surface states/fluxes. So we did not consider such variations of root water uptake depth. To address this limitation, we have added discussion on the shortage of this model into the discussion part. Please find L18-19 on P23. In addition, the soil layer depth refers to the maximum root water uptake depth, which can vary with time (Guderle and Hildebrandt, 2015), but SVEN simplified this soil depth parameter to keep it consistent. Guderle, M. and Hildebrandt, A.: Using measured soil water contents to estimate evapotranspiration and root water uptake profiles-a comparative study, Hydrol. Earth Syst. Sci., doi:10.5194/hess-19-409-2015, 2015.

16. Page 13, Line 7-9, Supplement Table S3 Please add the values for each of the initial conditions.

Reply: Thank you for your suggestion. We have added the values for the initial conditions into Table S3.

Table S3. Information on model initial conditions Initial conditions Description Unit Initial value CWSin Initial canopy water storage m 0 SWSin Initial soil water storage m 0.5 Ts0 Initial surface temperature âĎĊ Ta Td0 Initial deep soil temperature âĎĊ Ta

17. Results Page 15, Section 4.1 Not sure whether I overlooked this, but can you please indicate the values of the calibrated parameters? Also: I like Fig 4 showing the objective function. Near the pareto optimum plot a number of potentially very good model runs. Are they all roughly similar parameter values or do they differ substantially? This would give an indication of how well defined this model is in terms of the processes that are represented or/and the sensitivity of some of the parameters. Can you comment on this?

Reply: Thank you for your suggestion. The values of calibrated parameters are shown in L25-26 on P16. But to make it clearer, we also added the calibrated values of parameters directly to the figure caption (L6 on P17). Regarding whether optimized parameter values are similar or different, we have added the analysis on the optimized parameter values in supplementary Figure S1. Cveg and SWSmax show low variation of coefficients (CVs), and this indicates the parsimony of the SVEN model. Meanwhile, Csat and b show relatively higher CVs. This may be due to equifinality between Csat and b, which relate to soil thermal properties (Eq. 8) and could compensate each other.

18. Page 15 Lines 19-20, Page 18, Lines 16-20. I feel the numbers are crowding the text, and are difficult to take in. It is enough to refer to Fig 5, Fig 7 or alternatively collect them in a Table.

Reply: Thank you for your suggestion. We have streamlined these texts. Here we only put the performance regarding RMSDs in the text. Other statistic indices have been moved to Table 3 and Figure 8.

19. Page 16, Line 5, Line 8 To me Ts does not appear to be underestimated only in high NDVI conditions. Ts is also underestimated in May, when GPP is still very low. I am not convinced of this distinction .. but in order to support your point, you could color the points in the top right panel of Fig 5 with shades indicating NDVI (or GPP).

Reply: Thank you for your suggestion. We have revised Figure 7 to be the boxplot showing the simulation residuals and NDVI. We have also improved the interpretation

[Figure]

Interactive
comment

of results. We agree that the model tends to overestimate Ts for most cases. For details, please refer to L14-16 on P17.

20. Page 16, Line 24, Fig 5 Would be good to indicate the times of the seven snapshots in Fig 5 by vertical lines (solid for all UAS, dashed for augmented with tower data), so it is easier to see when the data was obtained for calibration.

Reply: Thank you for your suggestion. We have revised Fig 6 (original Fig 5) by adding vertical lines to show the UAS observations. Please see L5 on P18 in the revised version.

21. Page 16, Line 28 Do you mean "nearby" instead of "nearly"?

Reply: Thank you for your suggestion. We have revised this sentence. Please see L16 on P19 in the revised version.

22. Page 18, Line 1 I think "be" should be erased

Reply: Thank you for your suggestion. We have erased "be" and revised this sentence.

23. Page 18, Line 4-5, Fig 6 Can you please indicate in Fig 6 what the red lines refer to? I am at a loss, especially in panel (a). Also, I am not sure how the conclusion "GPP was underestimated under diffuse radiation conditions" is seen from the Figure, I am assuming in panel (j). Does the point cloud show a trend?

Reply: Thank you for your suggestion. The red lines in Fig 6 refers to that the model simulation residuals are equal to 0. To make this clear, we have added detailed explanation to the caption. Please see L4 on P20. In addition, we have revised the original scatter plots to the boxplots, which could be clearer to identify how the model simulation performance changes with NDVI and radiation conditions.

24. Page 18, Line 6 Add "of" after enhancement

Reply: Thank you for your suggestion. We have revised this sentence to make it clear. Please see L7 on P20 in the revised version.

25. Fig 7 Fonts in the top and bottom panels are not the same. Fig 5 & Fig 7 I was confused at first about the difference of the Fig 7 to the right panels in Fig 5. I concluded they are the same, just showing different time intervals. Can you collect them in one Fig? It would be easier to compare.

Reply: Thank you for your suggestion. To make figures clear, we have revised the figure to make fonts consistent. In addition, we have merged Fig 7 and Fig 5. Please see L1-6 on P21 in the revised version.

---

## Author Comment (AC3) · 30 Mar 2020

Response to the review of Referee 3. We have copied the comments of the referee hereunder with our comments appearing after the referee's comments.

1. General comments "Temporal interpolation of land surface fluxes derived from remote sensing – results with an Unmanned Aerial System" describes the use of a suite of simple models to interpolate surface fluxes and surface state variables between sporadically available land surface measurements. A model dubbed the SVEN was created by augmenting a Priestly Taylor model with new components to enable its use at timescales as short as 30-min. Instantaneous remotely sensed variables recorded

mainly using a UAS were used to calibrate the model, and the model was then used to fill in the extensive gaps between measurements. This work introduces and demonstrates of the technique, which was designed to be used with both satellite and UAS remote sensing measurements. It is a solid manuscript, with room for some improvement. Because one of the primary stated goals of the paper is the development of an application to satellite remote sensing, the omission of actual satellite measurements is conspicuous. I suggest that more attention be given to the topic of using satellite data. For example, what might be the shortcomings of applying the model to satellite-based measurements? Were UAS measurements relied upon for this paper rather than space-based measurements due to the inadequate spatial resolution of satellite measurements? At this site in-situ and UAS measurements are available (and used), but how well will the model work for the rest of the Earth's surface? Clarify the purpose of the model (including the parameter fitting) in the broadest sense, and develop, test, and describe the results accordingly. In addition, more care needs to be taken with the way remotely sensed measurements are handled. They are misleadingly referred to as "ground truth" or direct measurements throughout the manuscript, when most of the variables derived from remote sensing data were modeled or inferred, rather than measured directly. Uncertainties due to this also require more attention. The writing should be reviewed carefully by a native English speaker. Some examples are included in the specific comments below, but the manuscript includes many errors in writing and sentence structure.

Reply: We appreciate the reviewer's insightful comments and suggestions, which were very helpful to improve the manuscript. We totally agree that the great potential of utilizing the simple but effective land surface models to fill gaps between observed surface states and fluxes from remote sensing. We have thoroughly revised the manuscript to improve the presentation of this work. We have also added the discussion on the shortcomings of applying this model to satellite-based measurements and the rest of the Earth's surface. For instance, the SVEN model is a very simple water balance model, which has limited capacity to simulate soil water dynamics particularly in regions with complex landforms. In our study, SVEN also achieved moderate performance to simulate soil water dynamics. In addition, the soil layer depth refers to the maximum root water uptake depth, which may vary with time, but the model simplified this soil depth parameter to keep it consistent. Thus, in our study, SVEN also achieved moderate performance to simulate soil water dynamics. Meanwhile, the PT-JPL model has the limited performance to simulate ET in the dryland regions. There also remain challenges to get the reliability of atmospheric forcing such as radiation, precipitation and wind speed, particularly for data-scarcity regions. Moreover, the remote sensing based estimates of land surface temperature and soil moisture have uncertainties, which could be propagated to induce significant errors in the simulated continuous land surface variables. In addition, satellite based observations or estimates can have large uncertainties due to the coarse resolution. The integration of land surface model and satellite earth observation might be challenging than the integration with UAS derived variables. Please find the details in the discussion 4.4 (P22-23). We have also revised the words on "ground truth". We have changed the words to the UAS derived observations or estimates. We have also thoroughly revised the language and improve the manuscript writing. Here, we have also addressed your comments point-by-point.

2. Specific comments P 1, ln 7-8. With the exception of Ts, all of these variables are determined using remote sensing products based on a suite of different models and assumptions. For example, different vegetation indices can be measured remotely, but GPP cannot. The same applies to ET, SM, and Rn – none of these variables are measured directly using remote sensing, but the first sentence misleadingly indicates otherwise. Without detracting from main point of this sentence, a word such as "inferred" or "derived" could easily be included for more accuracy.

Reply: Thank you for your suggestion. We have revised the terminology to use "derived" to indicate that variables such as GPP, ET, SM ($\theta$) and Rn were estimated from remote sensing data.

3. P 1, ln 20. Delete the word, "well" from, "...SVEN can well estimate...". Awkward

as written.

Reply: Thank you for your suggestion. We have deleted this word. Please see L20 on P1 in the revised clean version.

4. P 2, ln 2-3. "Minimum parameterization" is awkward as written.

Reply: Thank you for your suggestion. We have deleted these words. Please see L1 on P2.

5. P 2, ln 7. "Mostly needed" is awkward as written. Also, replace "high frequency" with "prevalence."

Reply: Thank you. We have deleted "mostly" and have replaced "high frequency" with "prevalence". Please see L6 on P2 in the revised clean version.

6. P 2, ln 11. Replace "flexibly" and "favorable" with more precise descriptors.

Reply: Thank you. We have revised "flexibly" with "favorably". Please see L10 on P2.

7. P 2, ln 14. Replace "still just provide" with "still only provide."

Reply: Thank you. We have revised "still just provide" with "still only provide". Please see L20 on P2.

8. P 2, ln 16. Replace "uncovered" with "unknown."

Reply: Thank you. We have revised "uncovered" with "unknown". Please see L22 on P2.

9. P 2, ln 25. "using statistical interpolation could be challenging. . ." is awkward as written.

Reply: Thank you. We have revised this sentence to be "the statistical method to interpolate for variables that change substantially at sub-daily or daily time scales in response to the surface energy dynamics, e.g. Ts, Rn, SM, ET and GPP, could be challenging". Please see L30 on P2.

10. P 2, ln 28-29. "can be better" is awkward as written.

Reply: Thank you. We have revised this sentence to be "has great potential". Please see L1-2 on P3.

11. P 2, ln 30. Delete "a" in, "in a variable climate conditions."

Reply: Thank you. We have deleted "a".

12. P 3, ln 6. "as for example the turbulent fluxes are typically. . ." is awkward as written.

Reply: Thank you. We have deleted the sentence.

13. P 3, ln 7. "simpler but operational models based interpolation" is awkward as written.

Reply: Thank you. We have revised the sentence to be "Simple model based interpolation can be utilized to interpolate snapshot remote sensing estimates of land surface variables." Please see L12-13 on P3.

14. P 3, ln 11. Delete "the" at the beginning of this line.

Reply: Thank you. We have deleted that. Please see L14 on P3 in the revised version.

15. P 3, ln 15. Rewrite as, "limited meteorological inputs, and parameters that. . .".

Reply: Thank you for your suggestion. We have revised this sentence to be "We aimed at using prescribed vegetation dynamics from EO based vegetation indices, limited meteorological inputs, and parameters optimized from remote sensing derived fluxes to estimate temporally continuous land surface variables". Please see L19-21 on P3.

16. P3, ln 21-22. "now becomes" is awkward as written.

Reply: Thank you. We have revised the words to be "serve as". Please see L27 on P3 in the revised clean version.

17. P4, ln 4. Add the word, "it" after "forcing".
Reply: Thank you. We have added "it" after "forcing". Please see L10 on P4.

18. P4, In 17. Change, "onboard have been conducted" to, "onboard were conducted." And "Details refer…" to, "For more details refer...".

Reply: Thank you. We have changed the sentence to be "were conducted" and "for details, please refer to". Please see L22-23 on P4.

19. Figure 1. This is exactly the same as Figure 1 from Wang et al. (2018b). I don't know HESS's rules regarding this type of thing, so I will refer to the Editor for guidance. I would never reuse a figure like this myself, but if this is actually acceptable, the original usage should certainly be referenced. In addition, a small wind rose would be a nice addition to the figure; at a measurement height of 10 m (Wang et al., 2018b), the flux footprint will extend well beyond the edges of the figure in some conditions. As an aside to be passed onto the site manager, if the eddy covariance instrumentation were closer to the top of the canopy, it would help alleviate this problem."

Reply: Thank you for the comment. We don't want to reuse a figure from another paper. There are several differences between this figure and the one in Wang et al. (2018b). For instance, the new figure used the pseudo-color multispectral imagery (Red: 800 nm, Green: 670 nm, Blue: 530 nm) as the base map, while the one in another paper used a normal RGB photo as the base map. The new figure does not have markers to indicate the samples of soil moisture as Wang et al. (2018b). However, we admit that these two figures are similar. We have revised the figure to have more differences with Wang et al. (2018b). For instance, we have added the wind rose to Figure 1. Regarding the measurement height, Wang et al. (2018b) did not use $CO_2$ and water vapor eddy covariance data and the measurement height of 10 m refers to the meteorological observations such as wind speed, solar radiation, and longwave radiation. To make this clear, we have added more explanation to the data section. The $CO_2$ and water vapor eddy covariance system was adjusted to around 2 m above the maximum canopy height. This means that 2 m (before willow growing) to maximum 4
m (maximum canopy height of 6 m) above zero plane displacement. So, in most cases except for few stable conditions in night, the footprints of eddy covariance did extend beyond the edge of willow plantation. For details, please see L10-15 on P5.

20. P5, ln 24. "Data of few UAS flight campaigns" is awkward as written.

Reply: Thank you for the comment. We have revised this sentence to be "UAS data on June 24th were missing as shown in Table 1". Please see L10 on P6.

21. P5, ln 26. Replace the word, "resemble" with more appropriate verbiage. Reply: Thank you. We have changed the word to be "simulate". Please see L12 on P6.

22. P5, ln 27. Clarify that the "ground truth" SM measurements were not actual SM measurements, and describe the uncertainty and shortcomings of the remotely sensed SM product in detail.

Reply: Thank you for your suggestion. We have revised the "ground truth" words. For model calibration, the instantaneous values of the Ts and $\theta$ estimated from the seven UAS flights were used as reference.

23. P5, ln 29. "which corresponded to the willow emerging period with a high growth rate" is awkward as written.

Reply: Thank you for your suggestion. We have revised this sentence. The minimum revisit time was 10 days in the willow emerging period between May 2nd and May 12th. Please see L14-15 on P6.

24. P7, ln3. "and can facilitates to temporally interpolate" is awkward as written.

Reply: Thank you for your suggestion. We have revised the sentence to "can temporally interpolate the instantaneous land surface variables". Please see L11 on P7.

25. P8, ln 24-26. Clarify that this includes the existence of a canopy. As written, it reads like a simple soil diffusion-based approach, that neglects the existence of vegetation. There is a transfer coefficient for the canopy (Cveg) described on P9, along with LE

etc., so I assume this all adds up correctly (I am not a modeler), but a more complete initial description is wanted on P8.

Reply: Thank you for the comment. This model is proposed to simulate GPP and components of evapotranspiration (transpiration, evaporation from the intercepted water, and soil evaporation). Therefore, this model has the vegetation module to calculate the heat exchange between vegetation and ground. To make this clear, we have added more explanations at the beginning of the model description. Please see L16-17 on P7 in the revised clean version.

26. P9, ln 24-25. Change to, "k is the von Karman constant."

Reply: Thank you. We have revised this sentence. Please see L13 on P10.

27. P12, ln 20. "The rest of constraints," is awkward as written.

Reply: Thank you. We have revised this sentence. Other constraints such as thermal regulation reflect changes in LUE due to environmental factors. Please see L9-10 on P13.

28. P12, ln 21. "are the same modifying," is awkward as written.

Reply: Thank you. We have revised this sentence. "are the same for regulating ETc" Please see L10 on P13.

29. P13, ln 6. Change to, "UAS-derived observations," or otherwise clarify that many of these UAS variables were not measured directly.

Reply: Thank you. We have revised the sentence. The model inputs of this study were obtained from meteorological data, UAS derived observations or estimates.

30. P14, ln 1. Change, "facilitate," to, "facilitates."

Reply: Thank you. We have revised the word. Please see L8 on P15.

31. Eq 34 description. Clarify what time period (e.g. 30 min or 24 h) was used for this

[Figure]

EC measurement adjustment, and how missing data were handled.

Reply: Thank you. The EC data energy balance closure errors were corrected at 30 mins using the Bowen ratio approach. We have elaborated this in L18-19 on P15. Regarding the missing data, the data gaps were filled with based on the R-package REddyProc (Wutzler et al., 2018) using the meteorological data as inputs. For details, please refer to L13-15 on P5.

32. P14, ln 25. "well represent" is awkward as written. Reply: Thank you. We have revised the words to be "the indicators of". Please see L9 on P16.

33. Validation at the daily time scale Section. Augment the discussion of uncertainty in the UAS-derived measurements (as compared to direct measurements).

Reply: Thank you. We have added the discussion on the uncertainties of UAS derived estimates compared to the direct measurements. Please see L9 on P19.

34. P16, ln 24. "that the better UAS based snapshot estimates of SM..." is awkward as written. Perhaps, "that improving the UAS-based estimates of SM...".

Reply: Thank you for the comment. We have revised this sentence. Please see L13 on P19.

35. P16, ln 34. "has a large coverage" is awkward as written. More accurately, it could be replaced with something like: "extended well-beyond the edges of the Willow forest of interest".

Reply: Thank you. We have revised this sentence. During the night time, the eddy covariance footprint extended well-beyond the edges of the willow forest of interest, due to the stable atmospheric conditions. Please see L22 on P19.

36. P18, ln 1. "are be good" is awkward as written.

Reply: Thank you. We have revised this sentence to be "To check the model simulation performance under cloudy conditions". Please see L6 on P20.

37. P18, ln 3. "do not show difference" is awkward as written.

Reply: Thank you. We have revised this sentence. There were no significant differences for the residuals of the simulated Ts, Rn, SM and LE under low and high diffuse radiation fraction conditions. Please see L7-8 on P20.

38. P18, ln 5. "do to that the model" is awkward as written.

Reply: Thank you. We have revised this sentence to make it clearer. Please see L6-10 on P20.

39. P18, ln 6. "enhancement diffuse radiation effects" is awkward as written.

Reply: Thank you. We have revised this sentence. Please see L6-10 on P20.

40. P18, ln 19. Perhaps change, "R2 for Ts..." to, "R2 for monthly Ts..."

Reply: Thank you. We have revised the sentence to be monthly Ts.

41. P20, ln 10. Change, "understanding on the" to, "understand of the".

Reply: Thank you. We have revised the words. Please see L7 on P24.

42. Conclusions. What would the effects of using space-based remote sensing measurements be, rather that UAS measurements? Also discuss how well this method will work in areas where in-situ measurements are unavailable to better parameterize the UAS and SVEN measurements.

Reply: Thank you for your comments. We think there would be four major effects of using space-borne remote sensing measurements rather than UAS measurements. First of all, the space-borne remote sensing data have much coarser spatial resolution. If we move the simulation to the large scale with satellite data, we need to find accurate gridded meteorological data as forcing. UAS imagery has limited coverage and thus this study only used one meteorological station data as forcing. As satellite data have coarser pixel sizes, we also need to consider the sub-grid heterogeneity and identify the

effective values for model parameters. Note all parameter values of models were obtained from parameter calibration with remote sensing based estimates. For instance, in our study, we used the look-up tables with soil texture information to identify soil parameter values. In the large-scale simulation with satellite data, the plant functional type and soil type parameterization scheme for different ecosystems and environmental conditions would be needed. However, the integration of accurate remote sensing estimates with land surface models would be beneficial to reduce the dependency of plant functional type parameterization scheme and achieve a higher accuracy to predict land surface variables. In addition, coarse resolution satellite data may have limited accuracy to predict land surface fluxes compared to the detailed UAS data. Applying SVEN with satellite data to large scale, we also need to be careful about the accuracy of remote sensing based estimates and the error propagation from the model inputs to the outputs. Satellite data in the optical and thermal ranges can only provide observations during the sunny weather conditions. However, the UAS data in this study were collected in both sunny and cloudy conditions. We envision that using satellite based data to calibrate model may lead the model estimates biased towards the sunny conditions. Regarding the second question on applying this method in areas where in-situ measurements are unavailable to parameterize SVEN, there could be challenges to get reliable estimates of land surface fluxes. As shown in the model implementation of Fig. 4, the major challenges in data-scarcity regions would be lacking meteorological inputs and soil information. Such meteorological variables could be obtained from the online weather forecasting, although these data might not be as good as the standard weather station measurements. Soil parameters (e.g. hydraulic conductance, soil wilting point, and saturated soil moisture) could be obtained from soil texture maps or using model calibration with remote sensing based soil moisture estimates. For example, the soil moisture with high frequency across the entire growing season could be very helpful to identify the soil wilting point and saturated soil moisture, which could be close to the minimum and maximum values of soil moisture time series respectively. We also admit that estimating land surface fluxes in the data-scarcity regions is challenging and our

proposed approach could potentially have more uncertainties compared to the performance in the regions with rich in-situ measurements. But we believe using the remote sensing data from satellites or UAS can facilitate the prediction of land surface fluxes in data-scarcity regions. We want to keep the conclusion streamlined, so we have added the content into the discussion part to address the potential challenges for applying such methodology to satellite data and data-scarcity regions. Please see L8-34 on P23 in the revised version.

43. Equation and variable abbreviations. I cheated and read the other Reviewer Comments. I disagree with Referee #2 regarding their objection to the use of multiple letter abbreviations; I am already familiar with LUE, PAR, GPP, ET, etc., so their usage made it easier for me to follow the manuscript. In addition (this may have more to with my background than what is most suitable for HESS), I am more accustomed to Ïť than SM for soil moisture, and R (surface runoff) could easily be confused for respiration (although honestly I am not sure if there is a more widely used abbreviation for runoff).

Reply: Thank you for your suggestions. We agree that using abbreviations such as LUE, PAR, GPP and ET would be better for the readers. We have revised some variable abbreviations to make them easier for readers. For instance, in the revised version, we have used Qs to stand for surface runoff. We have used $\theta$ to represent soil moisture. Furthermore, we have also summarized all abbreviations in the supplementary material.

———————————————

---

## Referee Report (RR1)

**Comments on the revised version of manuscript hess-2019-0409**

**-1- Comments on runoff generation and leakage**
From the text, it is not clear (at least to me) the type of mechanism for runoff generation simulated by the SVEN model. From my knowledge of the water balance bucketing approach as well as what written in this paper, it seems that SVEN can simulate only the Dunne runoff generation (i.e. the saturation-excess mechanism), but not the Hortonian mechanism. Please, clarify that point in the text to the benefit of a wider readership.

As for the computation of the leakage losses (or drainage rate; see my Comment #2), it is not clear from the text how these losses become active. Commonly, in a bucket hydrological model, the leakage losses occur only when the actual soil moisture content is greater than soil moisture at the condition of "field capacity". Please, clarify this point.

Table 2 reports the "soil wilting point" (although it is placed in a wrong row, I guess), but it is not clear where this variable comes into play in the model. Did the authors make a correspondence between the "residual soil moisture" and "soil wilting point"? However, these two soil moisture contents have completely different meaning and clarification is required.

**-2- Comment on the usage of the term "percolation"**
I understand that the term "percolation" is sometimes employed in the literature when describing a downward flux in a soil water balance model. However, when addressing the downward flux at the lowest boundary of the soil domain in a bucket-type model such that used in this paper, a more common term is "drainage" or "leakage".
I would suggest the authors should think about changing the term "percolation". Some reasons for suggesting this change are below:
− In most cases, percolation is used when a groundwater table is present in the flow domain;
− More recently, the term "percolation" is associated with the so-called "percolation theory" that is employed to model the permeability of saturated rocks, or to determine percolation thresholds. In a lesser extent, "percolation" is referred to the downward water flow in natural soils toward the groundwater (Ghanbarian and Hunt, 2017, Geoderma 303:9-18);
− Some authors prefer using the term "percolation" to account for the fact that depth variations in downward flows are also caused by lateral flow, besides the temporal variations in precipitation and evapotranspiration, but it does not seem to me that the SVEN model accounts for later fluxes..

At P.12, L.17, I disagree with this statement. I suggest writing as follows: percolation (or drainage, or leakage) "… is computed by assuming the condition of a unit gradient of the total hydraulic potential [i.e. $\nabla(z+\psi)=-1$] at the lowest boundary and using …".

**-3- Comments on Eqs. (30)**
I do not think that the reference to the Mualem (1976) paper is relevant to Eq.(30). Mualem developed the well-known integral form of the unsaturated hydraulic conductivity and only exploit the Brooks-Corey soil-water retention function to link the two relationships of the soil hydraulic properties.
Instead, Eq.(30) was developed in the renowned van Genuchten (1980) paper. This reference is reported below:
van Genuchten, M.T., 1980. A closed-form equation for predicting the hydraulic conductivity of unsaturated soils. Soil Science Society of America Journal 44(5):892–898.

At P.12, L.22, I suggest changing the sentence as follows: "… *n* is a fitting parameter depending on the pore-size distribution. …"

**-4- Comments on Eq.(32) and variable SWS**

Variable SWS should be clearly defined in the text.

If I understood well, SWS is the product of soil moisture ($\theta$) and the soil depth ($Z$). Is that correct?

If yes, then Eq.(32) can be read as follows: $SWS/SWS_{max} = (\theta \times Z)/(\theta_s \times Z)$

Therefore, I do not think that this equation is really required in the paper.

Moreover, at P.16, L.25-26, the authors reported a calibrated value for $SWS_{max}$ equal to 554.52 mm (i.e. about 0.55 m). From Table 2, $\theta_s$ ranges from 0.38 to 0.43 $m^3m^{-3}$. Does it mean that the soil depth $Z$ of your soil profile ranges from about 1.46 m to about 1.29 m (i.e. $Z = SWS_{max}/\theta_s$)?

If the above is correct, it seems a quite deep soil profile, doesn't it? Since $SWS_{max}$ is a calibrated value, I think that "soil depth" starts losing its physical meaning. Please, clarify that point to the benefit of a wider readership.

**-5- Comments on Table 2**

5.1) I suggest the author should always refer to $K_s$ as the saturated hydraulic conductivity (not sometimes as the infiltration rate at soil saturation). Please, chose only one definition for a variable and use it throughout the manuscript. Moreover, from Table 3 of Dettmann et al. (2014), the range of $K_s$ is from 0.05 mm/h to $50.0\times10^3$ mm/h. Please, clarify the range you used in your model.

5.2) $n$ is the parameter of the van Genuchen (1980) (vG) hydraulic conductivity relation $K(\theta_e)$. Actually, it is the shape parameter of the vG soil-water retention relationship.

5.3) I have a suspicion that the last two variables of this table have been shifted. Moreover, there is a mistake because the same symbol $\theta_s$ was used to address both the "soil wilting point" and "saturated soil moisture". Moreover, it does not seem to me that the paper by Carsel and Parrish (1988) reports values for soil moisture at the permanent wilting point. Please check.

**-6- Comments on units of variables**

Please be consistent with S.I. units of measurement. For example, SWS is in units of "m" (see also the Supplement document and its Table S3), therefore I suggest that in the main manuscript one writes $SWS_{max}=0.55452$ m (instead of 554.52 mm). In all cases, non-S.I. units can be put in parenthesis, if required.

Moreover, it is also good practice in a scientific paper to use the same number of digits. Therefore, one may write, for example, $SWS_{max}=0.554$ m ($=5.54\times10^{-1}$ m).

---

## Author Response (AR2)

Dear Dr. Romano,

We would like to thank you and another reviewer for the insightful comments and suggestions to improve this manuscript. We have carefully revised the manuscript and have addressed all reviewers' comments one-by-one accordingly. Specifically, we have revised this manuscript in the following major points. (1) We have added the discussion on the reason for using the Dunne runoff generation process and the possibility to use the Hortonian approach when applying this model to other regions. (2) We have changed the terminology of "percolation" to "soil water drainage". (3) We have revised the wrong references on computing soil water drainage and the parameter calibration range (Table 2).

Our responses to all reviewers' comments are in the section below. The original reviewer comments are in black and our responses are in red.

Thank you so much!

Sincerely yours,

Sheng Wang
On behalf of all authors

Reviewer 1:

-1- Comments on runoff generation and leakage From the text, it is not clear (at least to me) the type of mechanism for runoff generation simulated by the SVEN model. From my knowledge of the water balance bucketing approach as well as what written in this paper, it seems that SVEN can simulate only the Dunne runoff generation (i.e. the saturation excess mechanism), but not the Hortonian mechanism. Please, clarify that point in the text to the benefit of a wider readership.

Reply: Thank you for the comments. In this study, we used the Dunne runoff mechanism (runoff occurs after saturation) due to two reasons. First of all, this bioenergy forest study site has a humid climate and the soil is quite thick. In most days around the year, the rainfall rate is low (Fig. 1) and trees can further reduce the effective rainfall rate due to interception. Furthermore, the terrain of this site is quite flat and the loamy soil has also a relatively large saturated hydraulic conductivity. Thus, the major mechanism for runoff generation in this site is Dunne runoff generation, which overland flow occurs after the saturation of soil water. In addition, the scope of this study was on proposing a simple and parsimonious approach to interpolate the remote sensing based land surface water, energy and $CO_2$ variables into the temporally continuous record. This model tried to use the simplest approaches to simulate surface energy, water and $CO_2$ fluxes. Thus, the current model only considered the Dunne mechanism and did not use the Hortonian mechanism. However, we agree that when applying this model to other regions with different soil, vegetation and topographic conditions, the model needs to be adapted to the Hortonian mechanism to simulate the runoff generation. We have added the contents in the discussion part to elaborate on the possibility to use the Hortonian runoff generation processes when applying this model to other regions. Please find the details on P19 L9-11 and L17 of the clean version manuscript.

Dunne, T. and Black, R. D.: An Experimental Investigation of Runoff Production in Permeable Soils, Water Resour. Res., doi:10.1029/WR006i002p00478, 1970.

Horton, R. E.: The Role of infiltration in the hydrologic cycle, Eos, Trans. Am. Geophys. Union, doi:10.1029/TR014i001p00446, 1933.

As for the computation of the leakage losses (or drainage rate; see my Comment #2), it is not clear from the text how these losses become active. Commonly, in a bucket hydrological model, the leakage losses occur only when the actual soil moisture content is greater than soil moisture at the condition of "field capacity". Please, clarify this point.

Reply: Thank you for the comments. We have changed the terminology of "percolation" to "soil water drainage". Here, the soil water drainage refers to water loss due to the gravity and can be estimated by Eq. (30), which assume the condition of a unit gradient of the total hydraulic potential at the lowest boundary and using the van Genuchten (1980) soil-water retention relationship. For details, please refer to P12 L17-19.

van Genuchten, M. T.: A Closed-form Equation for Predicting the Hydraulic Conductivity of Unsaturated Soils, Soil Sci. Soc. Am. J., doi:10.2136/sssaj1980.03615995004400050002x, 1980.

Table 2 reports the "soil wilting point" (although it is placed in a wrong row, I guess), but it is not clear where this variable comes into play in the model. Did the authors make a correspondence between the "residual soil moisture" and "soil wilting point"? However, these two soil moisture contents have completely different meaning and clarification is required.

Reply: Thank you for the comments. We have revised the wrong row. Because this study aims to propose a simple and parsimonious model to interpolate and simulate surface energy, water, and $CO_2$ fluxes, the model did simplification to assume that the residual soil moisture is equal to soil wilting point. We have added contents and explanations in the discussion part to address this simplification. For details, please refer to L8-9 on P19.

-2- Comment on the usage of the term "percolation" I understand that the term "percolation" is sometimes employed in the literature when describing a downward flux in a soil water balance model. However, when addressing the downward flux at the lowest boundary of the soil domain in a bucket-type model such that used in this paper, a more common term is "drainage" or "leakage".

I would suggest the authors should think about changing the term "percolation". Some reasons for suggesting this change are below:

− In most cases, percolation is used when a groundwater table is present in the flow domain;

− More recently, the term "percolation" is associated with the so-called "percolation theory" that is employed to model the permeability of saturated rocks, or to determine percolation thresholds. In a lesser extent, "percolation" is referred to the downward water flow in natural soils toward the groundwater (Ghanbarian and Hunt, 2017, Geoderma 303:9-18);

− Some authors prefer using the term "percolation" to account for the fact that depth variations in downward flows are also caused by lateral flow, besides the temporal variations in precipitation and evapotranspiration, but it does not seem to me that the SVEN model accounts for later fluxes..

At P.12, L.17, I disagree with this statement. I suggest writing as follows: percolation (or drainage, or leakage) "… is computed by assuming the condition of a unit gradient of the total hydraulic potential [i.e. $\nabla(z+\psi)=-1$] at the lowest boundary and using …".

Reply: Thank you for the comments. We agree that percolation should be used for groundwater and this word is not suitable for this context. We have revised the terminology according to your suggestions. We have revised the term on "percolation" to "drainage", according to Romano et al. (2011).

We have also revised the sentence at P12, L17. Soil water drainage, which is leakage out of the lower boundary of the flow domain (Romano et al., 2011), is computed by assuming the condition of a unit gradient of the total hydraulic potential at the lowest boundary and using the van Genuchten (1980) soil-water retention relationship as Eq. (30).

Romano, N., M. Palladino, and G. B. Chirico. "Parameterization of a bucket model for soil-vegetation-atmosphere modeling under seasonal climatic regimes." Hydrology & Earth System Sciences 15.12 (2011).

-3- Comments on Eqs. (30) I do not think that the reference to the Mualem (1976) paper is relevant to Eq.(30). Mualem developed the well-known integral form of the unsaturated hydraulic conductivity and only exploit the BrooksCorey soil-water retention function to link the two relationships of the soil hydraulic properties.

Instead, Eq.(30) was developed in the renowned van Genuchten (1980) paper. This reference is reported below:

van Genuchten, M.T., 1980. A closed-form equation for predicting the hydraulic conductivity of unsaturated soils. Soil Science Society of America Journal 44(5):892–898.

Reply: Thank you for the comments. We have revised the literature and corrected the explanations for Eq. 30. For details, please refer to P12, L18-19.

At P.12, L.22, I suggest changing the sentence as follows: "… n is a fitting parameter depending on the pore-size distribution. …"

Reply: Thank you for the comments. We have also revised this sentence. n is the shape parameter of the van Genuchen (1980) soil-water retention relationship, and depends on the pore-size distribution. For details, please refer to P12, L22-23.

-4- Comments on Eq.(32) and variable SWS

Variable SWS should be clearly defined in the text. If I understood well, SWS is the product of soil moisture ($\theta$) and the soil depth (Z). Is that correct?

Reply: Thank you for the comments. Yes, SWS is the product of soil moisture and the soil depth.

If yes, then Eq.(32) can be read as follows: $SWS/SWSmax = (\theta \times Z)/(\theta s \times Z)$

Therefore, I do not think that this equation is really required in the paper.

Reply: To streamline the manuscript, we have deleted Eq. (32) in the paper.

Moreover, at P.16, L.25-26, the authors reported a calibrated value for SWSmax equal to 554.52 mm (i.e. about 0.55 m). From Table 2, $\theta s$ ranges from 0.38 to 0.43 m3m-3 . Does it mean that the soil depth Z of your soil profile ranges from about 1.46 m to about 1.29 m (i.e. Z = SWSmax/ $\theta s$)?

If the above is correct, it seems a quite deep soil profile, doesn't it? Since SWSmax is a calibrated value, I think that "soil depth" starts losing its physical meaning. Please, clarify that point to the benefit of a wider readership.

Reply: Yes, the calculation of the soil profile is right. In this bioenergy willow forest site, the soil profile is quite deep and can reach to 1.4 m. But we also agree that this SVEN model is quite simple and $SWS_{max}$ is a calibrated parameter value, which may be different from the actual physical conditions. We have added

the contents on explaining the calibrated parameter values in the discussion part. For details, please refer to L1-2 on P17.

-5- Comments on Table 2

5.1) I suggest the author should always refer to Ks as the saturated hydraulic conductivity (not sometimes as the infiltration rate at soil saturation). Please, chose only one definition for a variable and use it throughout the manuscript. Moreover, from Table 3 of Dettmann et al. (2014), the range of Ks is from 0.05 mm/h to 50.0×103 mm/h. Please, clarify the range you used in your model.

Reply: Thank you for your comments. We have revised $K_s$ as the saturated hydraulic conductivity. We have checked the range of Ks and it should be from 0.05 mm/h to $50.0 \times 10^3$ mm/h.

Dettmann, U., Bechtold, M., Frahm, E. and Tiemeyer, B.: On the applicability of unimodal and bimodal van Genuchten-Mualem based models to peat and other organic soils under evaporation conditions, J. Hydrol., 515, 103–115, doi:10.1016/j.jhydrol.2014.04.047, 2014.

5.2) n is the parameter of the van Genuchen (1980) (vG) hydraulic conductivity relation K(θe). Actually, it is the shape parameter of the vG soil-water retention relationship.

Reply: Thank you for your comments. We have revised this part, according to your suggestions. For details, please refer to Table 2.

5.3) I have a suspicion that the last two variables of this table have been shifted. Moreover, there is a mistake because the same symbol θs was used to address both the "soil wilting point" and "saturated soil moisture". Moreover, it does not seem to me that the paper by Carsel and Parrish (1988) reports values for soil moisture at the permanent wilting point. Please check.

Reply: Thank you for your comments. We have revised the mistake on the shifting of the last two variables in Table 2. θs refers to saturated soil moisture, while $\theta_r$ represents the residual soil moisture. In the model, we have simplification to assume that soil wilting point is equal to the residual soil moisture. We have carefully checked the range for θs and θr in Carsel and Parrish (1988). For all soil types, $\theta_s$ should be [0.36, 0.46] and $\theta_r$ is [0.034,0.100].

Carsel, R. F. and Parrish, R. S.: Developing joint probability distributions of soil water retention characteristics, Water Resour. Res., 24(5), 755–769, doi:10.1029/WR024i005p00755, 1988.

-6- Comments on units of variables Please be consistent with S.I. units of measurement. For example, SWS is in units of "m" (see also the Supplement document and its Table S3), therefore I suggest that in the main

manuscript one writes $SWS_{max}=0.55452$ m (instead of 554.52 mm). In all cases, non-S.I. units can be put in parenthesis, if required.

Moreover, it is also good practice in a scientific paper to use the same number of digits. Therefore, one may write, for example, $SWS_{max}=0.554$ m ($=5.54\times10^{-1}$m).

Reply: Thank you for your suggestions. We have revised the unit of $SWS_{max}$ to S.I. units. $SWS_{max} = 5.54\times10^{-1}$ m. We have also checked units for other variables and changed all non-S.I. units to S.I. units.

Reviewer 2:

New Fig. 4: I like this new figure! Could you state the abbreviations "LUT" and "cal" in the figure caption. Also, could you add a very short note on the color coding of the boxes (text is stated).

Reply: Thank you so much for the suggestion. We have revised the caption to add explanations on the abbreviations and the note on the color coding of the boxes. The "LUT-based" refers to that the SVEN parameter values were obtained from look-up tables. The "Cal-based" indicates that the SVEN parameter values were acquired through calibration with UAS based surface temperature ($T_s$) and soil moisture ($\theta$). The red color box refers to 
[revised manuscript text omitted]